# Outlier Gradient Analysis:
# Efficiently Identifying Detrimental Training Samples for Deep Learning Models

**Anshuman Chhabra** [1]   **Bo Li** [2]   **Jian Chen** [2]   **Prasant Mohapatra** [1]   **Hongfu Liu** [3]

## Abstract

A core data-centric learning challenge is the identification of training samples that are detrimental to model performance. Influence functions serve as a prominent tool for this task and offer a robust framework for assessing training data influence on model predictions. Despite their widespread use, their high computational cost associated with calculating the inverse of the Hessian matrix pose constraints, particularly when analyzing large-sized deep models. In this paper, we establish a bridge between identifying detrimental training samples via influence functions and outlier gradient detection. This transformation not only presents a straightforward and Hessian-free formulation but also provides insights into the role of the gradient in sample impact. Through systematic empirical evaluations, we first validate the hypothesis of our proposed outlier gradient analysis approach on synthetic datasets. We then demonstrate its effectiveness in detecting mislabeled samples in vision models and selecting data samples for improving performance of natural language processing transformer models. We also extend its use to influential sample identification for fine-tuning Large Language Models.

## 1. Introduction

*Data-centric* learning focuses on enhancing algorithmic performance from the perspective of the training data (Oala et al., 2023). In contrast to *model-centric* learning, which designs novel algorithms or optimization techniques for performance improvement with fixed training data, data-centric learning operates with a fixed learning algorithm while modifying the training data through trimming, augmenting,

or other processing for improving utility (Zha et al., 2023). Data-centric learning holds significant potential in many areas such as model interpretation, subset training set selection, data generation, noisy label detection, active learning, and others (Chhabra et al., 2024; Kwon et al., 2024).

The essence of data-centric learning lies in estimating *data influence*, also known as data valuation in the context of a learning task (Hammoudeh & Lowd, 2022). Intuitively, the impact of an individual data sample can be measured by assessing the change in learning utility when training with and without that specific sample. This *leave-one-out* influence (Cook & Weisberg, 1982) provides a rough gauge of the relative data influence of the specific sample on the otherwise full fixed training set. *Shapley value* (Ghorbani & Zou, 2019; Jia et al., 2019), originating from cooperative game theory, quantifies the increase in value when a group of samples collaborates to achieve the learning goal. Unlike leave-one-out influence, Shapley value represents the weighted average utility change resulting from adding the sample to different training subsets. Despite the absence of assumptions on the learning model, the aforementioned retraining-based methods incur significant computational costs, especially for large-scale data analysis and deep models (Schioppa et al., 2022).

A popular choice for data valuation applications, such as identifying training samples detrimental to model performance, are *influence functions* (Koh & Liang, 2017). Essentially, influence functions assess data influence without requiring model retraining. They measure the effect of changing an infinitesimal weight of training samples based on a utility-evaluating function. While influence functions can be accurate or acceptable proxies for convex and certain shallow models, their applicability to deep models is constrained by the strong convexity assumption and the computational cost linked to calculating the inverse of the Hessian matrix (Basu et al., 2020a).

**Our Contributions**. In this paper, we delve into the classical data-centric problem: *identifying/trimming detrimental samples*. We tackle the computational challenge of the inverse of the Hessian matrix in influence functions in the context of detrimental sample identification and removal. Our major contributions are as follows:

---

[1]University of South Florida, Tampa, FL, USA [2]Tsinghua University [3]Brandeis University, Waltham, MA, USA. Correspondence to: Hongfu Liu <hongfuliu@brandeis.edu>.

*Proceedings of the $42^{nd}$ International Conference on Machine Learning*, Vancouver, Canada. PMLR 267, 2025. Copyright 2025 by the author(s).

- We build a bridge between identifying detrimental training samples via influence functions and outlier detection on the gradient space of samples, and propose our outlier gradient analysis approach. The transformation features a straightforward and Hessian-free formulation, and reduces the computational cost associated with the Hessian matrix and its inverse.

- Empirically, we utilize both linear and non-linear synthetic datasets to illustrate the ineffectiveness of the current Hessian approximation and to validate our hypothesis regarding outlier gradient analysis, showcasing our method's high accuracy in identifying mislabeled detrimental samples.

- Subsequently, we demonstrate the effectiveness of outlier gradient analysis in trimming mislabeled samples from vision datasets across various noise regimes. Additionally, we explore textual applications on data selection for fine-tuning deep transformer models and identifying influential data for text generation tasks using fine-tuned Large Language Models.

## 2. Related Work

**Retraining-Based Influence Estimation.** Influence estimation approaches can be generally categorized as either *retraining-based* or *gradient-based* (Hammoudeh & Lowd, 2022). Retraining-based methods consist of the classical leave-one-out influence approach (Cook & Weisberg, 1982), which consists of removing one training sample at a time, and retraining the model to measure sample influence via performance change. Other representative methods include Shapley value approaches (Ghorbani & Zou, 2019; Jia et al., 2019; Kwon & Zou, 2022), which are model agnostic, but also computationally untenable for large datasets and deep models due to exponential time complexity. Computationally efficient approaches such as KNN-Shap (Jia et al., 2018) can only employ KNN classifiers and hence are not directly applicable to the deep models.

**Gradient-Based Influence Estimation.** For models trained using gradient descent, gradient-based influence approaches can be used to approximately estimate influence without requiring retraining. The seminal work in this category is that of (Koh & Liang, 2017), which utilizes a Taylor-series approximation and LiSSA optimization (Agarwal et al., 2017) to compute sample influences. However, the limiting underlying assumption in the formulation is that the model and loss function are convex, which is not true for deep models. Follow-up works such as representer point (Yeh et al., 2018) and Hydra (Chen et al., 2021) inherit these convexity assumptions and suffer from similar issues of applicability. While influence functions have been used for numerous applications in data-centric learning (Feldman &

Zhang, 2020; Chhabra et al., 2024; Richardson et al., 2023), they tend to be too computationally expensive for large models, and cannot run in reasonable time. More recently, efficient influence estimation methods such as DataInf (Kwon et al., 2024), Arnoldi iteration (Schioppa et al., 2022), and Kronecker-factored approximation curvature (Grosse et al., 2023) have been proposed which can be employed for large models. Some approaches simply consider the gradients directly as a measure of influence (Pruthi et al., 2020; Charpiat et al., 2019), followed by some ensemble strategies (Bae et al., 2024; Kim et al., 2024). Recent work has also investigated the role of the Hessian and convexity in influence estimation (Schioppa et al., 2024). In contrast, our work aims to circumvent these issues for detrimental sample identification by operating on the gradient space in a skillful manner. Hence, our work paves the way for an efficient and accurate detrimental sample identification framework and adds to the "influence function toolset" for deep models and large datasets. Finally, recent work has also found that *self-influence* (influence computed on training samples) can be beneficial in detecting detrimental samples (Bejan et al., 2023; Thakkar et al., 2023). For related works on miscellaneous data-centric learning, please refer to Appendix A.

## 3. Proposed Approach

We first introduce influence functions conceptually and outline how they are applied to the task of detrimental samples identification. We then detail our transformation by converting the original formulation into a gradient space outlier analysis problem. Subsequently, we provide insights for extending influence functions to non-convex learning models and propose our *outlier gradient analysis* approach.

### 3.1. Preliminaries on Influence Functions

Let $T=\{z_i\}_{i=1}^n$ be a training set, where $z_i =(x_i, y_i)$ includes the input space feature $x_i$ and output space label $y_i$. A classifier trained using empirical risk minimization on the empirical loss $\ell$ can be written as: $\hat{\theta}=\arg\min_{\theta\in\Theta} \frac{1}{n}\sum_{i=1}^n \ell(z_i;\theta)$. Influence functions (Cook & Weisberg, 1982; Hampel, 1974; Martin & Yohai, 1986) measure the effect of changing an infinitesimal weight of training samples, based on a function that evaluates model utility. Downweighting a training sample $z_j$ by a very small fraction $\epsilon$ leads to a model parameter: $\hat{\theta}(z_j; -\epsilon) = \arg\min_{\theta\in\Theta} \frac{1}{n}(\sum_{i=1}^n \ell(z_i;\theta)-\epsilon\ell(z_j;\theta))$. By evaluating the limit as $\epsilon$ approaches 1, the seminal work of Koh & Liang (2017) provides an estimation for the *influence score* associated with the removal of $z_j$ from the training set in terms of training/validation loss as follows:

$$\mathcal{I}(z_j) = - \sum_{z\in T/V} \nabla_{\hat{\theta}}\ell(z;\hat{\theta})^\top \mathbf{H}_{\hat{\theta}}^{-1}\nabla_{\hat{\theta}}\ell(z_j;\hat{\theta}), \quad (1)$$

where $T/V$ denotes the training/validation set, $\nabla_{\hat{\theta}} \ell(z_j; \hat{\theta})$ is the gradient of the loss with respect to network parameters, and $\mathbf{H}_{\hat{\theta}} = \sum_{i=1}^{n} \nabla_{\hat{\theta}}^2 \ell(z_i; \hat{\theta})$ denotes the Hessian matrix.

One key application of influence functions lies in identifying detrimental samples. This is because an intuitive way of assessing whether a sample is detrimental is by training the model both with and without the specific training sample and computing metrics like training/validation loss. In other words, if the performance improves when excluding a particular sample, it is deemed detrimental to the learning task. By computing the influence score without needing to retrain the model, one can estimate the impact of a sample to assess if it is beneficial or detrimental, as follows:

$$\tilde{\mathcal{I}}(z_j) = \begin{cases} 0 \text{ (Detrimental Sample)} & \mathcal{I}(z_j) < 0. \\ 1 \text{ (Beneficial Sample)} & \mathcal{I}(z_j) \geq 0. \end{cases} \quad (2)$$

$\tilde{\mathcal{I}}(z_j)$ can be regarded as the discrete version of $\mathcal{I}(z_j)$. Specifically, a value of 0 for $\tilde{\mathcal{I}}(z_j)$ means that removing the sample $z_j$ enhances the model's utility, and that $z_j$ is a detrimental sample.

**Remark.** While influence functions offer a swift estimation for identifying detrimental training samples without the need for costly model retraining, their practical applications to large models are constrained by two prominent drawbacks. The first limitation lies in the necessity of a strictly convex loss function to guarantee the existence of the inverse of the Hessian matrix. The second challenge pertains to the considerable computational expense associated with calculating the inverse of the Hessian. For the first challenge, several possible solutions have been proposed: (1) a convex surrogate model can be used instead of the non-convex model (Chhabra et al., 2024); (2) a damping term can be added to the Hessian to ensure it is positive definite and invertible (Han et al., 2020); and (3) alternative formulations (Basu et al., 2020b; Alaa & Van Der Schaar, 2020) can be used (e.g. the Gauss Newton Hessian (Grosse et al., 2023) instead of the standard Hessian). Note that some studies bypass the convexity assumption and directly apply influence functions to deep models, yielding effective results. (Grosse et al., 2023). For the second challenge, various matrix inverse techniques are employed to expedite the computation process, including LiSSA optimization (Koh & Liang, 2017) and swapping the order of the matrix inversion (Kwon et al., 2024), among several others. Considerable efforts have been dedicated to addressing the aforementioned challenges with promising results– however, in this paper we target the second challenge for identifying/removing detrimental samples.

### 3.2. Bridging Influence Estimation and Outlier Analysis

We transform the problem of identifying detrimental samples via influence estimation to an outlier analysis problem in the gradient space. Upon scrutinizing the influence estimation of $z_j$ in Eq. (1), it becomes evident that the influence score is the result of three terms, with the first two remaining the same across all training samples and not solely dependent on $z_j$. While all three terms contribute to the concrete value of the influence score, it is the final term $\nabla_{\hat{\theta}} \ell(z_j; \hat{\theta})$ that assumes a decisive role in determining whether $z_j$ is a beneficial or detrimental sample. This is because the third term has $z_j$ as the only training sample as an input. With the following observation below regarding detrimental samples, we can build the connection between identifying detrimental samples via influence estimation and outlier analysis:

**Observation 3.1.** *For a converged model trained using empirical risk minimization, the majority of training samples positively contribute to the model's utility, and a much smaller subset than beneficial samples (with respect to the overall size of the training set) exhibits detrimental effects.*

Clearly, Observation 3.1 holds true as the empirical loss is an average of error between predictive and true values over all training samples. Hence, detrimental samples can be regarded as a minority outlier set compared to the beneficial sample majority. Based on Observation 3.1 and the decisive role of $\nabla_{\hat{\theta}} \ell(z_j; \hat{\theta})$ in influence estimation, we have the following hypothesis:

**Hypothesis 3.2.** *There exist outlier analysis algorithms capable of detecting detrimental samples in the gradient space. This algorithm would enable us to evaluate whether a training sample positively or negatively impacts model utility through influence estimation, effectively equating this evaluation with the application of the outlier analysis algorithm in the gradient space.*

Hypothesis 3.2 establishes a conceptual transformation between the identification of detrimental training samples via influence estimation and the detection of outliers in the gradient space. The outlying nature of detrimental samples has also been observed in past work (Kim et al., 2024). This transformation not only features a straightforward and Hessian-free formulation, reducing the computational cost associated with the Hessian matrix and its inverse, but also yields insights into the role of the gradient in sample impact beyond model optimization.

### 3.3. Our Approach: Outlier Gradient Analysis

As demonstrated in Hypothesis 3.2, outlier analysis can effectively be used to evaluate the discrete influence of training samples. Notably, we can circumvent the need for computing and inverting the Hessian for non-convex deep models by measuring discrete influence via Eq. (2). The primary contribution and discovery of our work lies in the realization that simple and efficient outlier analysis techniques can be applied to the gradient space for a discrete estimation of which samples are beneficial or

**Algorithm 1** : Outlier Gradient Analysis and Trimming

1: **Input**: Training set $T$, loss function $\ell$, model parameter $\hat{\theta}$, outlier analysis algorithm $\mathcal{A}$, trimming budget $k$
2: **Output**: Set $L$ containing beneficial/detrimental sample labels, Trimmed training set $T^*$
3: **initialize** $\mathcal{G} \leftarrow \emptyset, T^* \leftarrow \emptyset$.
4: $\mathcal{G} \leftarrow \mathcal{G} \cup \{\nabla_{\hat{\theta}}\ell(x_i, y_i; \hat{\theta})\}; \forall(x_i, y_i) \in T$.
5: $L \leftarrow \mathcal{A}(\mathcal{G}, k)$.
6: $T^* \leftarrow T^* \cup \{x_i\}; \forall L_i$ *is not an outlier*.
7: **return** $L, T^*$.

detrimental to the model's utility.

As Hypothesis 3.2 cannot prescribe a specific outlier detection algorithm, one of our choices for outlier analysis is the Isolation Forest (iForest) algorithm (Liu et al., 2008), owing to several factors. Firstly, iForest boasts a linear time complexity with a low constant, requiring minimal memory, rendering it well-suited for handling the high-dimensional gradient space inherent in deep models. Secondly, iForest constructs an ensemble of iTrees, where each iTree builds partial models and employs sub-sampling, demonstrating the ability to identify a suitable subspace for the detection of detrimental samples. Thirdly, iForest is known for its simplicity and effectiveness in identifying outliers that are non-linearly separated from inliers. Along with iForest, we also consider two simple outlier analysis approaches based on L1-norm and L2-norm thresholding, that work well in practice (Knorr et al., 2000).

Upon obtaining outlyingness labels through the application of an outlier detection algorithm to the gradient space, denoted as the set $L$, we can assess the influence of training samples on model performance. Subsequently, we then trim $k$ (the designated deletion budget) detrimental training samples. Retraining the model on this pruned sample set leads to potential performance improvements. The approach is outlined in Algorithm 1.

## 4. Hypothesis Verification on Synthetic Data

We seek to validate the hypothesis of our proposed idea and showcase the effectiveness of our outlier gradient analysis method on two synthetic 2D toy datasets[1] and two models for binary classification in Figure 1. In this figure, subfigures **A**-**D** present a linear dataset employing a Logistic Regression model, while subfigures **E**-**H** exhibit a non-linear dataset utilizing a non-convex Multilayer Perceptron (MLP) model as the base model. Specifically, subfigures **A** and **B** depict the training and test sets of a linearly separable dataset comprising 150 and 100 samples,

---

[1]Comprehensive details regarding datasets and model training for experiments are provided in Appendix B.

*Table 1.* Outlier detection and classification performance of noisy label correction and influence-based approaches including our proposed outlier gradient trimming on the *two half moons* dataset (top performer in bold).

| Method | Outlier Detection Accuracy (%) | Classification Post-Trimming (%) |
|---|---|---|
| Multilayer Perceptron | - | 90.0 |
| Normalized Margin | 82.0 | 89.0 |
| Self-Confidence | 82.0 | 89.0 |
| Confidence Entropy | 82.0 | 89.0 |
| Exact Hessian | 90.0 | 90.0 |
| Gradient Tracing | 82.0 | 91.0 |
| LiSSA | 82.0 | 91.0 |
| DataInf | 82.0 | 91.0 |
| Self-LiSSA | 82.0 | 90.0 |
| Self-DataInf | 90.0 | 87.0 |
| **Outlier Gradient (iForest)** | 96.0 | **96.0** |
| **Outlier Gradient (L1)** | **98.0** | 87.0 |
| **Outlier Gradient (L2)** | **98.0** | 87.0 |

respectively. Notably, the training set includes 10 manually generated noisy samples with misspecified labels. Subfigure **C** displays the influence score of each training sample, computed using Eq. (1), and subfigure **D** provides a visualization of the gradient space. Similarly, subfigures **E** and **F** represent the training and test sets of the *two half moons* dataset, with the training set consisting of 250 samples and the test set of 100 samples, equally distributed between two classes. The training set in this case also contains 20 noisy samples. Subfigures **G** and **H** showcase the influence score and gradient space of the non-convex case.

In the linear case, as illustrated in subfigure **C**, the influence score proves to be a reliable indicator for distinguishing detrimental samples from beneficial ones. Notably, detrimental samples exhibit large negative scores, while other samples display positive or nearly zero values. Additionally, subfigure **D** affirms that these detrimental samples are distinctly separated in the gradient space, confirming the validity of the equivalent transformation outlined in Hypothesis 3.2. However, the limitations of influence scores become evident in the context of non-convex models, as observed in subfigure **G**, where the influence scores of detrimental samples are mixed with those of normal ones. Nevertheless, in the gradient space illustrated in subfigure **H**, the detrimental samples are effectively isolated from inliers. Notably, our method does not rely on the Hessian for computing influence and operates directly on the gradient space using outlier analysis.

We also conduct a quantitative evaluation to assess the advantages of our approach compared to three recently proposed noisy label correction methods and six influence function-based approaches, as detailed in Table 1. Specifically, we measure ground-truth outlier predictive accuracy and the performance gain achieved by removing detrimental samples. For noisy label correction approaches we consider:

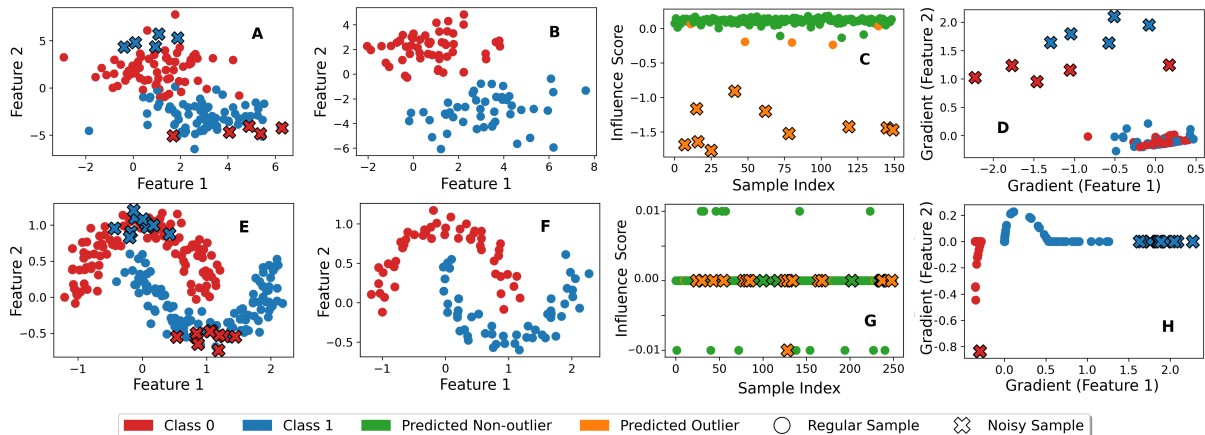

*Figure 1.* Illustrating our outlier gradient analysis approach on two synthetic datasets and convex/non-convex models. **A-D** showcase our outlier gradient analysis approach on a 2D linearly separable synthetic dataset. This dataset includes a small subset of detrimental samples with incorrect labels used to train a Logistic Regression binary classification model. Meanwhile, **E-H** depict our outlier gradient analysis on a non-linear synthetic dataset with mislabeled samples employed in training a Multilayer Perceptron (MLP) neural network. In subfigures **A** and **E**, the training sets are represented with class labels 0 (red) and 1 (blue) in the convex and non-convex cases, respectively. Detrimental samples with incorrect class labels are marked with ×, while regular samples are marked with ○. **B** and **F** denote the test sets used to evaluate model performance. **C** and **G** display the influence scores calculated by Eq. (1). Note that **G** demonstrates that influence scores are not reliable indicators for detecting detrimental samples in the non-convex case. After applying outlier analysis on the gradient space of the non-convex MLP model, most detrimental samples are detected. **D** and **H** showcase the gradient space obtained for each sample from the Logistic Regression and MLP models, respectively. It is evident that the outlier samples correspond to detrimental samples with mislabeled classes, which are linearly or non-linearly separated from inliers. Note that the benefits of outlier gradient trimming can be clearly observed—removing predicted outlier samples via iForest and retraining the MLP enhances classification performance from $90\% \rightarrow 96\%$ on the test set (refer to Table 1).

*Normalized Margin* (Northcutt et al., 2021), *Self-Confidence* (Müller & Markert, 2019), and *Confidence-Weighted Entropy* (Kuan & Mueller, 2022). The influence function approaches include computing the Hessian exactly (Cook & Weisberg, 1982), using the Hessian-free gradient tracing approach by (Pruthi et al., 2020), LiSSA-based optimization (Koh & Liang, 2017), the recently proposed influence estimation approach DataInf (Kwon et al., 2024), self-influence using LiSSA as in Bejan et al. (2023), and self-influence using DataInf. We compute influences only using the training samples and performance is measured on the test set.

Our outlier gradient analysis approaches demonstrate high accuracy in identifying mislabeled outliers (96-98%), outperforming all three noisy label correction baselines (only 82% accuracy) and among influence baselines, all exhibit similar performance except for exact Hessian computation, which attains 90% accuracy. Next, we evaluate model performance gain by removing detected outlier samples and retraining the MLP on the trimmed dataset. Here the benefits of our iForest outlier gradient analysis can be observed, as it increases performance from 90% to 96% while the overtly simple L1/L2-norm outlier analysis approaches are not as effective. The other baselines attain performance between 89-91%. This emphasizes the effectiveness of our iForest approach, while exhibiting low time complexity (refer to Appendix C.3 for details on computational complexity).

## 5. Noisy Label Correction for Vision Datasets

We now demonstrate the effectiveness of our approach in addressing noisy label correction using the *CIFAR-10N* and *CIFAR-100N* real-world noisy label datasets (Wei et al., 2022). These datasets stem from the original *CIFAR-10* and *CIFAR-100* datasets (Krizhevsky et al., 2009), but introduce label inaccuracies due to crowdsourced labeling. *CIFAR-10N* has 3 different noise settings: *Aggregate, Random,* and *Worst*– these correspond to using majority voting across 3 annotators, first annotator label, and worst annotator label, respectively. *CIFAR-100N* only has one noise setting.

Table 2 shows the accuracy performance of outlier gradient analysis (L1/L2-norm, iForest) compared to label correction approaches and influence-based baselines covered in the previous section. Exact Hessian computation is excluded due to its computational intractability for large datasets. Our outlier gradient analysis methods consistently outperform other baselines across diverse noise settings and datasets. Notably, even in challenging scenarios like the *Worst* noise setting in *CIFAR-10N* (40.21% noise rate), our approaches are the top performers– L1-norm based outlier analysis achieves highest accuracy gain, improving from 82.27% (vanilla ResNet-34) to 84.20%. Similar superior performance is observed in the *Random* noise setting (17.23% noise rate), where L2-norm outlier analysis achieves a final accuracy of 90.25% compared to original cross-entropy accuracy of 89.17% and

*Table 2.* Accuracy (5 runs) on *CIFAR-10N* and *CIFAR-100N* for a ResNet-34 model trained via cross entropy and performance post trimming using noisy label correction approaches and influence-based methods, including our outlier gradient analysis (top-2 performers in bold).

| Method | CIFAR-10N | | | CIFAR-100N |
|---|---|---|---|---|
| | Aggregate | Random | Worst | Noisy100 |
| Cross Entropy | 90.87 | 89.17 | 82.27 | 57.36 |
| Normalized Margin | 91.33 | 90.06 | 83.57 | 60.94 |
| Self-Confidence | 91.38 | 90.09 | 83.65 | 60.51 |
| Confidence Entropy | 91.11 | 90.05 | 83.63 | 60.62 |
| Gradient Tracing | 91.47 | 89.98 | 83.38 | 60.73 |
| LiSSA | 91.49 | 90.05 | 83.38 | 60.48 |
| DataInf | 91.46 | 90.05 | 83.40 | 60.70 |
| Self-LiSSA | **92.07** | 89.58 | 83.01 | 59.48 |
| Self-DataInf | 91.41 | 89.81 | 83.15 | 60.56 |
| **Outlier Gradient (L1)** | 91.86 | **90.66** | **84.20** | 60.32 |
| **Outlier Gradient (L2)** | **92.21** | 90.25 | 82.99 | **61.40** |
| **Outlier Gradient (iForest)** | 91.36 | 90.20 | **83.72** | **60.99** |

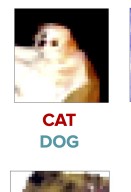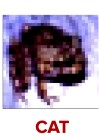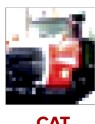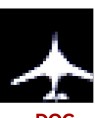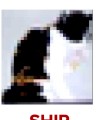

| CAT | CAT | CAT | DOG | SHIP |
|---|---|---|---|---|
| DOG | FROG | TRUCK | AIRPLANE | CAT |

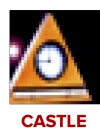

| LEOPARD | CASTLE | HOUSE | MOTORBIKE | CAR |
|---|---|---|---|---|
| KANGAROO | CLOCK | TREE | TULIP | FOX |

*Figure 2. Detrimental* samples detected using our outlier gradient analysis. Top row: *CIFAR-10N*; bottom row: *CIFAR-100N*. Top label (**red**): noisy label; bottom label (**green**): correct class.

in *CIFAR-100N*, where it attains the highest performance of 61.40%, surpassing the cross-entropy performance of 57.36%. In the *CIFAR-10N Aggregate* noise setting (noise rate 9.03%), outlier gradient analysis is again the top performer. Due to space constraints, we omit standard deviations from Table 2, but these are provided in Appendix C.1.

Additionally, visual examples of mislabeled samples detected by our outlier gradient analysis approach (iForest) are provided in Figure 2. All displayed images contain mislabeled samples, and their removal from the training set contributes to improved model performance on the test set. In Table 2, we set the trimming budget for outlier gradient analysis ($k$) at 5% of the training data size. An empirical analysis for the choice of $k$ is undertaken in Appendix C.2, where we vary the outlier budget (from 2.5% to 12.5%) and measure test set accuracy across the *CIFAR-10N* dataset.

**Additional Analyses.** We conduct ablations on the iForest parameters in Appendix C.4. Further, we provide running time experiments on *CIFAR-10N* and *CIFAR-100N* in Appendix C.3 along with the other baselines. We also provide results with ResNet-18 as the base model in Appendix C.5 and on ImageNet (Deng et al., 2009) in Appendix C.6,

showing similar trends. Finally, approaches for noisy learning can be categorized into methods that either change the loss function or model architecture or methods that identify noisy samples and remove/relabel them for improving performance (Algan & Ulusoy, 2021). Since our approach belongs to the latter category, we only compare against other approaches from this category. For completeness, we also present results comparing our approach with some others in the former category in Appendix C.7. We would like to emphasize that this is not an exhaustive list of baselines and noisy learning by adjusting the loss/model is not the focus of our work (but detecting detrimental samples is). Moreover, our algorithm could also be combined with approaches from both categories for additional gains. Finally, we also conducted experiments using two new influence function methods: TRAK (Park et al., 2023) and GEX (Kim et al., 2024). While we were able to obtain results for *CIFAR-10N* (please refer to Appendix C.8 for results), both methods got out-of-memory errors on *CIFAR-100N* for the same experimental set-up as other influence methods. Given their shortcomings, we did not consider them for the other experiments.

## 6. Data Selection for Fine-tuning NLP Models

We conduct experiments on data selection for fine-tuning on NLP models, following the experimental setup by Kwon et al. (2024) for DataInf, where the RoBERTa transformer model (Liu et al., 2019) is fine-tuned on four binary GLUE datasets (Wang et al., 2018): *QNLI*, *SST2*, *QQP*, and *MRPC*. To assess if influence-based methods can enhance NLP model performance via Low Rank Adaptation (LoRA) (Hu et al., 2022) fine-tuning, Kwon et al. (2024) introduce noisy versions of all four datasets by flipping the binary label for 20% randomly chosen training data samples. The goal of the data selection task is to select the best representative subset of the training data so that performance is maximized on an unseen test set. Specifically, 70% of the most beneficial samples are selected according to each influence computation approach, and the model is fine-tuned for 10 epochs and rank of LoRA matrix is set to 4. Then, as the model trains over each epoch, performance is measured on the unseen test set. Clearly, for fairness, the sample influence is computed only using the training set, and the test set remains unknown until inference.

The results over three runs are presented in Figure 3 for all four GLUE datasets. We only show trends for iForest based outlier gradient analysis to aid visualization since performance is similar for the L1/L2-norm methods. It can be seen that our outlier gradient trimming approach markedly outperforms all other baselines– more specifically, outlier gradient analysis achieves the best test set results on *QNLI, SST2, QQP*, and on *MRPC*, Self-LiSSA (Bejan et al., 2023) and outlier gradient analysis are on par with each other. Despite this competitive performance, our outlier gradient

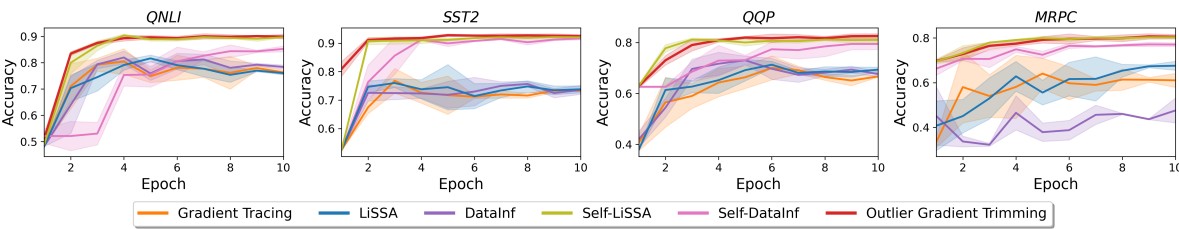

*Figure 3.* Performance of the data selection task using outlier gradient trimming and other influence baselines for fine-tuning RoBERTa.

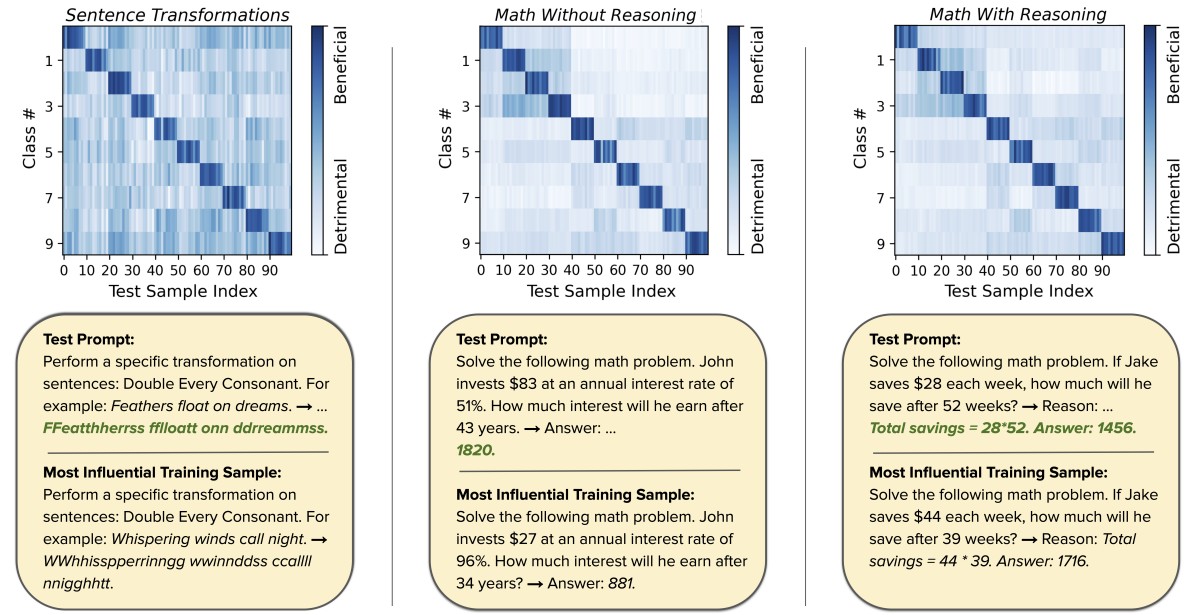

*Figure 4.* Results for outlier gradient analysis on LLM influential data identification benchmarks.

analysis is orders of magnitude faster than Self-LiSSA, as shown in experiments of Appendix C.3. This highlights the effectiveness of our proposed outlier gradient analysis approach in selecting relevant data for fine-tuning NLP models while being more computationally efficient.

# 7. Extending to Influential Data Identification for LLMs

We now consider an alternate task– demonstrating the effectiveness of our proposed outlier gradient analysis in identifying influential data samples for Large Language Models (LLMs), using the proposed benchmarks from DataInf (Kwon et al., 2024). The LLM influential data identification task at its core is a *similarity measurement* task, as it seeks to ascertain which fine-tuning prompts are most similar to a given test sample (Askari et al., 2025). More specifically, the goal is to assess what training set prompts (used for LoRA fine-tuning) are most influential for a given unseen test prompt. The robustness and effectiveness of influence estimation are gauged based on whether the identified training set prompts belong to the same class category as the given test prompt. We utilize the three benchmark datasets introduced in DataInf (Kwon et al., 2024): *Sen-*

*tence Transformations, Math Without Reasoning,* and *Math With Reasoning*, to conduct the influential data identification experiment on the *Llama-2-13B-chat* LLM (Touvron et al., 2023). For each of the influence identification benchmark datasets, there are 900 training samples for LoRA fine-tuning, and 10 categories or classes of task types with 90 samples belonging to each class. For each dataset there are 100 test set prompts with 10 test set prompts per class category.

In (Kwon et al., 2024), to predict the most influential training samples given a test set prompt, the authors assign a pseudo label to every data point in the training set (1 if it is in the same class/task category as the test data prompt, or 0 otherwise). This set serves as a ground-truth for measuring performance of identifying influential data samples. Next, they calculate the Area Under the Curve (AUC) by comparing the absolute values of the influence function (for each training set prompt corresponding to a given test prompt) with these pseudo labels. Clearly, a high AUC signifies that training data samples from the same category have a significant influence on the given test prompt. The average AUC across all test data points is then recorded, and is denoted as the Class Detection (AUC) metric. Additionally, another metric is used– for every test data prompt, the authors determine if the

*Table 3.* AUC/Recall for outlier gradient analysis and baselines for influential class detection for three tasks on *Llama2-13B* LLM.

| Task | Method | Class Detection (AUC) | Class Detection (Recall) |
|---|---|---|---|
| Sentence Transformations | Gradient Tracing | 0.999 ± 0.001 | 0.982 ± 0.032 |
| | DataInf | **1.000 ± 0.000** | 0.996 ± 0.012 |
| | **Outlier Gradient** | **1.000 ± 0.000** | **1.000 ± 0.000** |
| Math Problems Without Reasoning | Gradient Tracing | 0.724 ± 0.192 | 0.241 ± 0.385 |
| | DataInf | 0.999 ± 0.005 | 0.993 ± 0.046 |
| | **Outlier Gradient** | **1.000 ± 0.000** | **1.000 ± 0.000** |
| Math Problems With Reasoning | Gradient Tracing | 0.722 ± 0.192 | 0.226 ± 0.376 |
| | DataInf | 0.999 ± 0.004 | 0.990 ± 0.049 |
| | **Outlier Gradient** | **1.000 ± 0.000** | **1.000 ± 0.000** |

proportion of training data prompts belonging to the same class/category are within the top 90 (# of training prompts in each category) influential samples. The average % across all test data points is calculated and this metric is denoted as Class Detection (Recall), where higher recall is better.

As part of this task, we need to measure similarity between train and test set samples. Note that for our experiments on identifying detrimental samples outlier gradient analysis only operated on the training set (i.e., it uses the training set gradients). However, to extend outlier analysis to this task while maintaining consistency with the previous experiments and methods, we will train 10 individual iForest estimators for each class prompt category, as the ultimate objective is to use outlier gradient analysis for prompt class detection. Each class's iForest estimator is trained solely on the gradient space of training prompts from that category. Subsequently, for each test set prompt, we utilize each iForest estimator to generate an outlier score based on the gradient space of that test sample, enabling us to conduct the influential data identification experiment. Note that the other baseline influence methods already have access to the given test set sample and can use that information directly for analyzing which training sample is most influential.

Our outlier gradient analysis performs exceptionally well on this task, achieving perfect scores for both AUC and Recall in Table 3. It outperforms DataInf and Gradient Tracing, with LiSSA omitted as it fails to converge due to instability on LLMs (Kwon et al., 2024). Self-influence baselines also cannot be used since a similarity matrix with the full set of test prompts needs to be constructed (information leakage). Figure 4 further illustrates the individual influence predictions, with darker colors indicating lower outlier score magnitudes. The heatmaps correspond to three benchmark datasets, with test samples ordered sequentially based on their categories. The accurate influence estimation is evident from the highest influence values along the diagonal. The most influential sample identified by our approach closely resembles the given test prompts.

## 8. Discussion

**Computational complexity and running time.** Throughout, we have emphasized that outlier gradient analysis is efficient while being highly accurate at identifying detrimental training samples. We also conduct experiments to validate this empirically. In Table 6 (Appendix C.3), we benchmark the running time for all the methods considered for the various noise settings of *CIFAR-10N* and *CIFAR-100N*. It can be observed that outlier gradient analysis features in the top-performing methods in terms of computational efficiency, while simultaneously also featuring as a top-performing method for accurately detecting detrimental samples (as seen in Table 2). We observe similar trends for the ImageNet dataset in Table 10 (Appendix C.6). Note that this is also evident in terms of worst-case computational complexity, as outlier gradient analysis possesses linear (in both number of samples and parameters) time complexity (see Table 7 in Appendix C.3 for more details).

**Adapting outlier gradient analysis to a validation/test set distribution.** In some scenarios we might wish to utilize a validation set distribution to accurately adjust influence estimation. This is especially true for distribution shift scenarios, where the training and validation distributions are different. In the original influence formulation, the first term provides this information. For outlier gradient analysis, we only use training set gradients. To rectify this, we can instead employ a *semi-supervised* outlier analysis algorithm $\mathcal{A}$ with validation samples provided as *inliers*. We utilize the semi-supervised OneClassSVM (Li et al., 2003) outlier analysis algorithm and the distribution shift experimental framework from Chhabra et al. (2024) to assess performance. These results indicate that outlier gradient analysis is the top-performer across baselines, as can be seen in Table 13 (Appendix C.9). While a full extensive analysis of validation set adaptation is beyond the scope of this paper, these preliminary experiments showcase the benefits of outlier gradient analysis beyond just the training distribution.

## 9. Conclusion

We focused on the key data-centric learning task of identifying detrimental training samples. Influence functions are a leading approach often used for this problem, but possess certain deficiencies when applied to deep models, such as the computational demands for inverting the Hessian matrix. We propose a novel solution for detrimental sample detection that does not rely on the Hessian matrix, and hence eliminates this major limitation. Our approach, outlier gradient analysis, is based on a conceptual transformation between the influence function formulation and outlier analysis in the gradient space. This transformation results in a computationally efficient method that possesses high detection accuracy. Through comprehensive experiments on synthetic datasets and various application domains (code details in Appendix D), including noisy label correction for vision models, data selection for NLP models, and even influential data identification in LLMs, we demonstrated that

our method outperformed many existing influence-based approaches and baselines in deep learning scenarios.

## Impact Statement

Our work and proposed techniques aim to address the data-centric task of identifying detrimental samples. We improve upon the influence function analysis framework that is used to undertake this problem, but possesses deficiencies when applied to deep learning models. Enabling influence estimation for deep models allows practitioners to assess whether training samples are beneficial or detrimental to performance, and can make models more interpretable and performant. As we show through extensive experiments on multiple problem settings, our proposed outlier gradient analysis approach outperforms existing baselines and can augment model performance by identifying/trimming detrimental samples in a computationally efficient manner. As a result, our work paves the way for significant positive societal impact, especially with the increased adoption of larger and deeper neural networks such as LLMs. However, as with any work, there are limitations to our approaches that can be overcome in future work. For instance, it might be possible to derive specific outlier analysis algorithms that are computationally more efficient than iForest or norm thresholding, and significantly more performant. Another limitation that can be overcome is the further study and benchmarks for influence based analysis in LLMs– going beyond the datasets and approaches we used in this work. Further, while outlier gradient analysis is useful in cases where training data can be noisy, it might not be as useful if the data is very high quality and there are no outlying gradient samples. However, it is unlikely that this will be the case in the real-world unless some steps have been taken prior to training to ensure high data quality. Finally, outlier analysis algorithms have a fundamental limitation of how to specify the budget for outlier detection, which is a non-trivial hyperparameter optimization problem. While this is a common problem with little consensus across the entire field of outlier analysis, our methods inherit this limitation as well (although we note that outlier gradient analysis performs well for different budget thresholds, as shown in additional experiments in the Appendix C.2).

## Acknowledgments

The authors would like to thank Han Yue for aiding with experiment design and implementation, and the anonymous reviewers for their feedback in helping strengthen the work. Bo Li was supported by the National Natural Science Foundation of China (No. 72171131, 72133002). Anshuman Chhabra was supported by the USF CSE department faculty startup fund for the duration of this project.

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

# Appendix

# A. Additional Related Work on Miscellaneous Data-Centric Learning

Many works in the data-centric learning domain study other relevant research questions beyond detrimental sample identification and influence estimation. For instance, *datamodels* (Ilyas et al., 2022) also estimate training sample contributions, but only for one test sample at a time. *Data efficiency* approaches (Jain et al., 2023; Paul et al., 2021; Killamsetty et al., 2021) aim to accelerate deep learning training time via subset selection. *Data pruning* approaches based on novel approximations for leave-one-out influence estimation (Tan et al., 2024) and the model's generalization gap (Yang et al., 2022) have also been proposed. *Model pruning* via generalized influence functions has also been studied in (Lyu et al., 2023). Note that after identifying detrimental training samples, one can adopt multiple strategies for recourse. While we focus on removal in this paper, other alternatives could also be used, such as *relabeling* (Richardson et al., 2023; Kong et al., 2021). *Antidote data augmentation* (Chhabra et al., 2022; Li et al., 2023) methods aim to generate synthetic data samples to improve model performance, whereas *feature selection* approaches (Hall, 1999; Cai et al., 2018) seek to optimize the feature space to only those important for model performance. *Active learning* (Cohn et al., 1996) methods aim to iteratively identify optimal samples to annotate given a large unlabeled training data pool (Liu et al., 2021; Nguyen et al., 2022; Wei et al., 2015). Finally, works on *poisoning attacks* seek to analyze model robustness by perturbing training set samples (Solans et al., 2021; Mehrabi et al., 2021; Chhabra et al., 2023) under natural input constraints. The study of training sample influence has also been extended to recent generative models, such as diffusion models (Dai & Gifford, 2023), through the use of ensembles.

# B. Detailed Information on Datasets and Model Training

We describe dataset details as well as model training and other information used in the main paper.

## B.1. Datasets

We first cover our generated synthetic datasets, then the vision datasets– *CIFAR-10N* and *CIFAR-100N*, then provide more details on the four *GLUE* binary classification NLP datasets, and finally discuss details regarding the benchmark datasets for influential data identification in LLMs– *Sentence Transformations, Math Without Reasoning,* and *Math With Reasoning*.

### B.1.1. SYNTHETIC DATASETS

We conduct experiments for our proposed outlier gradient analysis and other baselines on two synthetic datasets. The first dataset is linearly separable for logistic regression classification and consists of 150 training samples and 100 test samples. These are created using the scikit-learn (Pedregosa et al., 2011) library's `make_blobs` function. For each of the two binary classes, we manually flip the labels of 10 samples (5 for each class) to add noise to the dataset. The second dataset is the non-linear *half moons* dataset so that we can train an MLP network with two hidden layers with ReLU activations. The training set has 250 samples and the test set has 100 samples, and the dataset is generated using the scikit-learn library's `make_moons` function. Here too, we manually flip the labels of 20 samples (10 from each class) to add noise to the data.

### B.1.2. *CIFAR-10N* AND *CIFAR-100N*

Both the *CIFAR-10N* and *CIFAR-100N* datasets (Wei et al., 2022) consist of the same input images that make up the *CIFAR-10* (10 classes) and *CIFAR-100* (100 classes) datasets (Krizhevsky et al., 2009), respectively. Each input is a 32x32 RGB image with dimension (3,32,32). However, for *CIFAR-10N* and *CIFAR-100N*, the labels are noisy, as they contain real-world human annotation errors collected using 3 annotators on Amazon Mechanical Turk. As these datasets are based on human-annotated noise, they model noisy real-world datasets more realistically, compared to synthetic data alternatives. The training set for both datasets contains 50,000 image-label pairs, and the test set contains 10,000 image-label pairs that are free from noise. For *CIFAR-10N* we utilize three noise settings for experiments in the paper– (1) *Worst*, which is the dataset version with the highest noise rate (40.21%) as the worst possible annotation label for the image is chosen, (2) *Aggregate*, which is the least noisy dataset (9.03%) as labels are chosen via majority voting amongst the annotations, and (3) *Random* which has intermediate noise (17.23%) and consists of picking one of the annotators' labels. We use the first annotator for the random labels. For *CIFAR-100N* there is only a single noisy setting (*Noisy100*) due to the large number of labeling classes, and the overall noise rate is 40.20%.

### B.1.3. *GLUE* DATASETS

The *GLUE* or the *General Language Understanding Evaluation* (Wang et al., 2018) benchmark datasets consist of a number of benchmarks for training, evaluating, and analyzing natural language models. As in the DataInf paper (Kwon et al., 2024), we utilize the four binary classification subset datasets: *QNLI, SST2, QQP,* and *MRPC* for experiments. Here, these datasets cover a wide variety of natural language task domains. For instance, *QNLI* (Wang et al., 2018) covers natural language inference, *SST2* (Socher et al., 2013) covers sentiment analysis, *QQP* [2] covers question answering, and *MRPC* (Dolan & Brockett, 2005) covers paraphrase detection. We use the same datasets as in Kwon et al. (2024), where the training and test splits are obtained from the Huggingface `datasets`[3] library. For *QQP* and *SST2* in Kwon et al. (2024) 4500 training samples and 500 test samples were randomly sampled from the full sets, so we utilize these in our experiments for a fair comparison.

### B.1.4. *Sentence Transformations*

For this benchmark dataset proposed in (Kwon et al., 2024), the LLM is required to perform a specific transformation on an input sentence. There are 10 different sentence transformations. To help the model learn different transformations, "chatbot" name identifiers are used and each is uniquely associated with each transformation. These are the categories of sentence transformations (taking an example input sentence as "Welcome to the real world."):

- **Reverse Order of Words**: world. real the to Welcome
- **Capitalize Every Other Letter**: wElCoMe To ThE rEaL wOrLd.
- **Insert Number 1 Between Every Word**: Welcome 1to 1the 1real 1world.
- **Replace Vowels with \***: W\*lc\*m\* t\* th\* r\*\*l w\*rld.
- **Double Every Consonant**: Wwellccomme tto tthhe rreall wworrlldd.
- **Capitalize Every Word**: Welcome To The Real World.
- **Remove All Vowels**: Wlcm t th rl wrld.
- **Add ly To End of Each Word**: Welcomely toly thely really world.ly
- **Remove All Consonants**: eoe o e ea o.
- **Repeat Each Word Twice**: Welcome Welcome to to the the real real world. world.

### B.1.5. *Math With/Without Reasoning*

Both these datasets consist of the same math problems that the LLM is tasked to solve, with the only difference being whether or not an intermediate reasoning step is used in prompting the model. More specifically the LLM is asked to provide a direct answer to an arithmetic math word problem. There are 10 types of word problems and random positive integers are used to construct unique prompts. These are as follows:

- **Pizza**: Jane ate A slices of pizza and her brother ate B slices from a pizza that originally had C slices. How many slices of the pizza are left? Reason: Combined slices eaten = A + B. Left = C - (A + B).
- **Chaperones**: For every A students going on a field trip, there are B adults needed as chaperones. If C students are attending, how many adults are needed? Reason: Adults needed = (B * C) // A.
- **Purchase**: In an aquarium, there are A sharks and B dolphins. If they bought C more sharks, how many sharks would be there in total? Reason: Total sharks = A + C.
- **Game**: John scored A points in the first game, B points in the second, C in the third, and D in the fourth game. What is his total points? Reason: Total points = A + B + C + D.
- **Reading**: Elise reads for A hours each day. How many hours does she read in total in B days? Reason: Total hours read = A * B.
- **Discount**: A shirt costs A. There's a B-dollar off sale. How much does the shirt cost after the discount? Reason: Cost after discount = A - B.
- **Area**: A rectangular garden has a length of A meters and a width of B meters. What is its area? Reason: Area = A * B.
- **Savings**: If James saves A each week, how much will he save after B weeks? Reason: Total savings = A * B.
- **Cupcakes**: A bakery sells cupcakes in boxes of A. If they have B cupcakes, how many boxes can they fill? Reason: Boxes filled = B // A.
- **Interest**: Jake invests A at an annual interest rate of B%. How much interest will he earn after C years? Reason:

---

[2]https://quoradata.quora.com/First-Quora-Dataset-Release-Question-Pairs
[3]https://huggingface.co/docs/datasets

Interest = (A * B * C) // 100.

## B.2. Models and Methods

We now describe the models and the methods used in our experiments throughout the main paper. First, we describe the ResNet-34 (He et al., 2016) architecture used as the base model for the noisy vision datasets, then the RoBERTa (Liu et al., 2019) NLP transformer model, and then the Llama-2 LLM.[4] We also describe implementation details and parameter values for the label correction baselines in Sections 4 and 5 and the influence-based baselines used throughout the paper. Finally, we also describe some key implementation details regarding our outlier gradient analysis approach.

### B.2.1. RESNET-34

The ResNet-34 model was proposed in (He et al., 2016) and is a 34-layer convolutional neural network pretrained on the ImageNet-1K dataset at resolution $224 \times 224$. The pretrained model block is fine-tuned on the *CIFAR-10N/CIFAR-100N* training set experiments with default parameters– minibatch size (128), optimizer (SGD), initial learning rate (0.1), momentum (0.9), weight decay (0.0005), and number of epochs (100), for all experiments. Moreover, we directly used the implementation provided by Wei et al. (2022) and made modifications to their code.

### B.2.2. ROBERTA

As in (Kwon et al., 2024), we utilize LoRA fine-tuning to fine-tune the RoBERTa-large model, a 355M parameter transformer language model that improves upon the original BERT model in key ways such as implementation and hyperparameter selection. LoRA is applied to every value matrix of the attention layers of the RoBERTa model. The pre-trained model from Huggingface is used.[5] A learning rate of 0.0003 and a batch size of 32 is used. The model is fine-tuned over 10 epochs using LoRA and dropout is set to be 0.05 while the rank of the LoRA matrix is set to 4, as recommended in Kwon et al. (2024). The loss function used is a negative log-likelihood as the datasets are all for binary classification. The LoRA training is enabled using the Huggingface PEFT library.[6] For the influence experiments we have utilized the code provided in (Kwon et al., 2024) and adapted it for our experiments. Moreover, we only compute influences using the training set gradients, and keep the test set hidden from the learning model for fair evaluation.

### B.2.3. LLAMA2-13B-CHAT LLM

We fine-tune the Llama2 13B parameter instruction tuned LLM using LoRA fine-tuning (applied to every query and value matrix of the attention layer) as in Kwon et al. (2024). The LoRA parameters are as follows: learning rate is set to be 0.0003, rank of LoRA matrix is set to 8, $\alpha = 32$ in 8-bit quantization, and the batch size is set to 32 across 25 fine-tuning epochs. A negative log-likelihood of the generated response is used as the loss function for fine-tuning as before. Here too, we adapt the code provided by Kwon et al. (2024) for our use cases.

### B.2.4. LABEL CORRECTION BASELINES

For label correction baselines in Sections 4 and 5– *Normalized Margin* (Northcutt et al., 2021), *Self-Confidence* (Müller & Markert, 2019), and *Confidence-Weighted Entropy* (Kuan & Mueller, 2022), we utilize the implementation provided in the Cleanlab[7] library. We use default parameters for all three baselines. Note that the baselines are model agnostic and only require predicted labels and associated probabilities for predictions, which we can easily obtain from classifiers.

### B.2.5. INFLUENCE-BASED BASELINES

We utilize three influence-based baselines in experiments: LiSSA (Koh & Liang, 2017), Gradient Tracing (Pruthi et al., 2020), DataInf (Kwon et al., 2024). For each of these baselines, we utilize the implementation provided in Kwon et al. (2024) and adapt it to our application scenarios. For each baseline influence estimation is undertaken only on the training set (except for additional results in adapting to the test set, provided in Appendix C.9 below). We only use the last checkpoint in Gradient Tracing (Pruthi et al., 2020) for fair comparisons.

---

[4]https://huggingface.co/meta-llama/Llama-2-13b-chat-hf.
[5]https://huggingface.co/docs/transformers/model_doc/roberta.
[6]https://huggingface.co/docs/peft/index.
[7]https://github.com/cleanlab/cleanlab/.

We now discuss implementation details regarding outlier gradient analysis. Owing to the simplicity of our approach, the implementation is straightforward and follows directly from the algorithm. In most cases, we directly utilize the gradients obtained from the last layer of the model being considered. However, in some cases, the gradient space of samples can be high dimensional. For instance, for *CIFAR-100N*, the gradient space is of dimension $50000 \times 51200$ which unnecessarily increases memory and time complexity of outlier detection. As a result, we reduce the gradient space dimensionality by employing a sparse random projection step (Li et al., 2006) where the reduced dimension is ascertained using the scikit-learn library. We also utilize sparse random projection in this manner for the *Llama-2-13B-chat* LLM model experiments to reduce the dimensionality of the gradient space obtained.

## C. Additional Results and Experiments

We now provide details on additional experiments. We first provide results for the noisy label datasets and vision models shown in the main paper, but with standard deviation included. Then we conduct ablation experiments on the outlier detection threshold $k$ for the outlier gradient analysis algorithm. We also provide experiments on running time of our proposed approach (as well as details on computational complexity), ablation experiments on varying iForest parameters, results on ImageNet, experiments with ResNet-18 as the base model instead of ResNet-34, among others.

*Table 4.* Accuracy ± Standard Deviation results obtained for 5 runs on the *CIFAR-10N* and *CIFAR-100N* datasets for a ResNet-34 model trained via cross entropy as well performance post trimming using noisy label correction approaches and influence-based methods, including our proposed outlier gradient analysis methods.

| Method | CIFAR-10N | | | CIFAR-100N |
| --- | --- | --- | --- | --- |
| | Aggregate | Random | Worst | Noisy100 |
| Cross Entropy | 90.87 ± 0.23 | 89.17 ± 0.31 | 82.27 ± 0.37 | 57.36 ± 0.43 |
| Normalized Margin (Northcutt et al., 2021) | 91.33 ± 0.11 | 90.06 ± 0.14 | 83.57 ± 0.32 | 60.94 ± 0.59 |
| Self-Confidence (Müller & Markert, 2019) | 91.38 ± 0.19 | 90.09 ± 0.17 | 83.65 ± 0.21 | 60.51 ± 0.51 |
| Confidence Entropy (Kuan & Mueller, 2022) | 91.11 ± 0.34 | 90.05 ± 0.26 | 83.63 ± 0.41 | 60.62 ± 0.26 |
| Gradient Tracing (Pruthi et al., 2020) | 91.47 ± 0.21 | 89.98 ± 0.20 | 83.38 ± 0.58 | 60.73 ± 0.38 |
| LiSSA (Koh & Liang, 2017) | 91.49 ± 0.34 | 90.05 ± 0.31 | 83.38 ± 0.58 | 60.48 ± 0.29 |
| DataInf (Kwon et al., 2024) | 91.46 ± 0.17 | 90.05 ± 0.38 | 83.40 ± 0.56 | 60.70 ± 0.31 |
| Self-LiSSA (Bejan et al., 2023) | **92.07 ± 0.15** | 89.58 ± 0.11 | 83.01 ± 0.34 | 59.48 ± 0.43 |
| Self-DataInf | 91.41 ± 0.17 | 89.81 ± 0.37 | 83.15 ± 0.22 | 60.56 ± 0.28 |
| **Outlier Gradient Analysis (L1)** | 91.86 ± 0.14 | **90.66 ± 0.33** | **84.20 ± 0.19** | 60.32 ± 0.42 |
| **Outlier Gradient Analysis (L2)** | **92.21 ± 0.14** | **90.25 ± 0.22** | 82.99 ± 0.54 | **61.40 ± 0.22** |
| **Outlier Gradient Analysis (iForest)** | 91.36 ± 0.09 | 90.20 ± 0.07 | **83.72 ± 0.18** | **60.99 ± 0.27** |

### C.1. Full Results with Standard Deviation for Vision Model Experiments

In the main paper results of Section 5 we provide accuracy values without the standard deviation listed, due to space constraints. Here, we augment those results by also providing the standard deviation obtained over the 5 runs. These results are denoted in Table 4. It can be seen that the standard deviations are in general low, and overall, outlier gradient trimming has low variance.

### C.2. Additional Results for Different Trimming Budget $k$

We now conduct experiments varying $k$ from 2.5% to 12.5% for all three noise settings and baselines in the *CIFAR-10N* dataset. These results are shown in Table 5. As can be observed, our outlier analysis approaches features in the top-2 irrespective of the value of $k$. Moreover, the highest values across each noise regime are obtained by outlier gradient analysis (L2 norm at 12.5% for *Aggregate* and *Random*; and L2 norm at 2.5% for *Worst*). Finally, we find that setting $k$ as 5% and 12.5% are good overall choices leading to consistently desirable performance. Hence, we select 5% as the outlier budget in experiments.

### C.3. Experiments on Running Time and Computational Complexity

We now present running time experiments for outlier gradient analysis on both the *CIFAR-10N* and *CIFAR-100N* datasets compared to the other baselines compared in the paper in Table 6. It can be seen that outlier gradient analysis is

*Table 5.* Varying the trimming budget $k$ and measuring test set performance across noisy datasets (top-2 performers at each $k$ in bold).

| CIFAR10N (Aggregate) | 2.5% | 5% | 7.5% | 10% | 12.5% |
|---|---|---|---|---|---|
| Gradient Tracing | **92.11** | 91.47 | **92.17** | 91.99 | 91.98 |
| LiSSA | 92.08 | 91.49 | 91.83 | 92.27 | 91.74 |
| DataInf | **92.34** | 91.46 | 91.81 | 91.80 | 92.07 |
| Self-LiSSA | 91.71 | **92.07** | 91.32 | 91.72 | 91.33 |
| Self-DataInf | 91.22 | 91.41 | 91.37 | 91.29 | 91.15 |
| Outlier Gradient (L1) | 91.39 | 91.86 | 92.05 | **92.36** | **92.21** |
| Outlier Gradient (L2) | 92.10 | **92.21** | **92.70** | **92.63** | **92.78** |
| Outlier Gradient (iForest) | 91.77 | 91.36 | 91.57 | 91.92 | 92.08 |
| **CIFAR10N (Random)** | **2.5%** | **5%** | **7.5%** | **10%** | **12.5%** |
| Gradient Tracing | **90.71** | 89.98 | 90.41 | **90.75** | 90.96 |
| LiSSA | 90.21 | 90.05 | **91.09** | **90.88** | 90.00 |
| DataInf | 90.77 | 90.05 | 90.30 | 90.26 | 90.80 |
| Self-LiSSA | 89.76 | 89.58 | 89.50 | 88.94 | 89.49 |
| Self-DataInf | 89.91 | 89.81 | 90.32 | 89.91 | 90.00 |
| Outlier Gradient (L1) | 90.51 | **90.66** | 90.24 | 90.45 | **91.17** |
| Outlier Gradient (L2) | **90.72** | **90.25** | **90.63** | 90.50 | **91.21** |
| Outlier Gradient (iForest) | 90.03 | 90.20 | 90.06 | 90.38 | 90.62 |
| **CIFAR10N (Worst)** | **2.5%** | **5%** | **7.5%** | **10%** | **12.5%** |
| Gradient Tracing | 83.56 | 83.38 | 83.61 | 84.12 | **84.49** |
| LiSSA | **84.51** | 83.38 | **84.25** | 83.63 | 83.89 |
| DataInf | 84.31 | **83.40** | 83.45 | 84.01 | 84.12 |
| Self-LiSSA | 82.65 | 83.01 | 82.75 | 82.71 | 82.66 |
| Self-DataInf | 83.70 | 83.15 | 83.53 | 82.96 | 83.84 |
| Outlier Gradient (L1) | 84.26 | **84.20** | 84.12 | 84.32 | 84.25 |
| Outlier Gradient (L2) | **84.48** | 82.99 | 84.09 | **84.35** | **84.43** |
| Outlier Gradient (iForest) | 83.74 | 83.72 | **84.22** | **84.44** | 83.25 |

computationally efficient and a fraction of the original running time of the model. Moreover, it is order of magnitudes faster than the other baselines. Thus, our outlier gradient analysis approach is computationally efficient as an option for trimming detrimental samples and improving model performance. Most notably, only Gradient Tracing is faster than outlier gradient analysis, but as we demonstrated in the main paper results, it seldom as accurate in detecting detrimental samples as outlier analysis. Thus, outlier gradient analysis is ideal for balancing performance with computational efficiency. We also provide analytical time complexity comparisons in Table 7. Although, it is important to note that in practice, outlier gradient analysis is much faster than the worst case time complexity, as can be seen in Table 6.

*Table 6.* Running time for our outlier gradient analysis approaches and other baselines (top-2 in bold).

| Method | Time Taken (seconds) | | | |
|---|---|---|---|---|
| | CIFAR-10N (Aggregate) | CIFAR-10N (Random) | CIFAR-10N (Worst) | CIFAR-100N (Noisy100) |
| Gradient Tracing | **0.30** | **0.30** | **0.39** | **5.45** |
| DataInf | 3.89 | 3.99 | 4.01 | 15.22 |
| LiSSA | 23.75 | 23.25 | 23.26 | 115.19 |
| Self-DataInf | 5.29 | 5.51 | 5.5 | 12.1 |
| Self-LiSSA | 30.44 | 31.64 | 31.07 | 94.93 |
| Outlier Gradient Analysis (L1) | **0.54** | **0.54** | **0.74** | 10.3 |
| Outlier Gradient Analysis (L2) | 0.55 | 0.55 | 0.8 | 8.99 |
| Outlier Gradient Analysis (iForest) | 2.09 | 2.15 | 2.19 | **8.46** |

## C.4. Experiments with Varying Tree Estimators

We conduct further ablations for our iForest outlier gradient analysis approach. The main parameter (other than the trimming budget $k$, which we investigate in Appendix C.2) of iForest based outlier gradient analysis is the number of tree estimators being used. As a result, we vary the number of these estimators, and measure performance. We observe that test set

*Table 7.* Computational complexity of outlier gradient analysis methods and other baseline approaches ($n$ is #training samples, $v$ is #validation/test samples, $p$ is #model parameters, $m$ is #inputs for LLM and $o$ is #outputs for LLM).

| Method | Type | Time Complexity |
|---|---|---|
| Exact (Eq 1) | Hessian-based | $\mathcal{O}(nv^3)$ |
| LiSSA (Koh & Liang, 2017) | Hessian-based | $\mathcal{O}(nvp)$ |
| DataInf (Kwon et al., 2024) | Hessian-based | $\mathcal{O}(nvp)$ |
| EK-FAC (Grosse et al., 2023) | Hessian-based | $\mathcal{O}(m^2o + p^2o)$ |
| Self-LiSSA (Bejan et al., 2023) | Self-influence | $\mathcal{O}(np)$ |
| Self-DataInf (Bejan et al., 2023) | Self-influence | $\mathcal{O}(np)$ |
| Gradient Tracing (Pruthi et al., 2020), | Hessian-free | $\mathcal{O}(nvp)$ |
| **Ours (Outlier Gradient Analysis)** | Hessian-free | $\mathcal{O}(np)$ |

performance on *CIFAR-10N* (*Worst* noise setting) for outlier gradient analysis remains stable across the board when the number of estimators are varied, as can be seen in Table 8.

*Table 8.* Results on varying the number of tree estimators used in iForest outlier gradient analysis.

| # Tree Estimators | 25 | 50 | 75 | 100 | 125 | 150 | 175 | 200 |
|---|---|---|---|---|---|---|---|---|
| Accuracy on Test Set (%) | 83.70 | 84.38 | 83.71 | 83.72 | 83.66 | 83.97 | 83.84 | 83.42 |

## C.5. Experiments on ResNet-18 Architecture

We also provide results for ResNet-18 (He et al., 2016) being used as the base model IN Table 9 instead of the ResNet-34 model. The overall performance of the ResNet-18 model is lower than ResNet-34 for all datasets and noise settings, since the ResNet-18 model has fewer residual connections than the ResNet-34 model. Moreover, it can be observed that outlier gradient analysis leads to improved performance post trimming, compared to the cross entropy baseline. Outlier gradient trimming is advantageous as a data selection strategy irrespective of the base model.

*Table 9.* Accuracy ± Standard Deviation results for 5 runs on the *CIFAR-10N* and *CIFAR-100N* datasets for a ResNet-18 model trained via cross entropy as well performance post trimming using noisy label correction approaches and our proposed outlier gradient analysis.

| Method | CIFAR-10N | | | CIFAR-100N |
|---|---|---|---|---|
| | *Aggregate* | *Random* | *Worst* | *Noisy100* |
| Cross Entropy | 90.78 ± 0.12 | 89.01 ± 0.31 | 81.85 ± 0.45 | 57.22 ± 0.12 |
| **Outlier Gradient Trimming (Ours)** | **91.17 ± 0.14** | **89.91 ± 0.21** | **83.08 ± 0.26** | **60.58 ± 0.28** |

## C.6. Experiments on ImageNet

Although noisy label experiments have not been conducted on ImageNet (Deng et al., 2009), we decided to undertake a simple experiment on a subset of ImageNet. We created a subset of ImageNet containing 50000 images (50 images from each of the 1000 classes) as the training set, and flipped 40% of the corresponding image labels to create noisy labels (20 images from each class). The validation set is the same as ImageNet with 50000 images. We obtain results for performance on this set for a baseline ResNet-18 (He et al., 2016) model, DataInf, Gradient Tracing, iForest based outlier gradient analysis, as well as simple L1-norm and L2-norm thresholding based outlier gradient analysis. The models are trained for 10 epochs. In this limited experimental setting, we obtain the following results in Table 10 and find that outlier gradient analysis methods achieve competitive performance to other methods while being highly computationally efficient.

## C.7. Experiments on Other Noisy Learning Baselines

As we discussed previously, approaches for noisy learning can be categorized into (1) methods that either change the loss function or model architecture or (2) those that identify noisy samples and remove/relabel them for improving model performance (Algan & Ulusoy, 2021). Since our approach belongs to the latter category, we only compared against other approaches from this category in the main paper. For completeness we now present results comparing our approach with

*Table 10.* Results on ImageNet (top-3 performers based on performance and time taken are in bold).

| Method | Accuracy (%) | Time Taken (s) |
|---|---|---|
| Cross Entropy | 49.2 | - |
| Gradient Tracing | 51.0 | **23.51** |
| DataInf | **51.5** | 182.3 |
| Outlier Gradient Analysis (iForest) | 50.3 | 103.5 |
| Outlier Gradient Analysis (L1) | **_51.5_** | **_44.81_** |
| Outlier Gradient Analysis (L2) | **51.2** | **44.68** |

*Table 11.* Comparing with the alternate category of noisy learning baselines.

| Method | CIFAR-10N (Aggregate) | CIFAR-10N (Random) | CIFAR-10N (Worst) |
|---|---|---|---|
| Backward-T (Patrini et al, 2017) | 88.13 ± 0.29 | 87.14 ± 0.34 | 77.61 ± 1.05 |
| Forward-T (Patrini et al, 2017) | 88.24 ± 0.22 | 86.88 ± 0.50 | 79.79 ± 0.46 |
| T-Revision (Xia et al, 2019) | 88.52 ± 0.17 | 88.33 ± 0.32 | 80.48 ± 1.20 |
| VolMinNet (Li et al, 2021) | 89.70 ± 0.21 | 88.30 ± 0.12 | 80.53 ± 0.20 |
| GCE (Zhang and Sabuncu, 2018) | 87.85 ± 0.70 | 87.61 ± 0.28 | 80.66 ± 0.35 |
| Peer Loss (Liu and Guo, 2020) | 90.75 ± 0.25 | 89.06 ± 0.11 | 82.00 ± 0.60 |
| F-Div (Wei and Liu, 2020) | 91.64 ± 0.34 | 89.70 ± 0.40 | 82.53 ± 0.52 |
| Positive-LS (Lukasik et al, 2020) | 91.57 ± 0.07 | 89.80 ± 0.28 | 82.76 ± 0.53 |
| Negative-LS (Wei et al, 2021) | 91.97 ± 0.46 | 90.29 ± 0.32 | 82.99 ± 0.36 |
| Co-teaching+ (Yu et al, 2019) | 90.61 ± 0.22 | 89.70 ± 0.27 | 83.26 ± 0.17 |
| JoCoR (Wei et al, 2020) | 91.44 ± 0.05 | 90.30 ± 0.20 | 83.37 ± 0.30 |
| ELR (Liu et al, 2020) | **92.38 ± 0.64** | **91.46 ± 0.38** | 83.58 ± 1.13 |
| CORES-2 (Cheng et al, 2020) | 91.23 ± 0.11 | 89.66 ± 0.32 | 83.60 ± 0.53 |
| **Outlier Gradient Analysis (L1)** | 91.86 ± 0.14 | **90.66 ± 0.33** | **84.20 ± 0.19** |
| **Outlier Gradient Analysis (L2)** | **92.21 ± 0.14** | 90.25 ± 0.22 | 82.99 ± 0.54 |
| **Outlier Gradient Analysis (iForest)** | 91.36 ± 0.09 | 90.20 ± 0.07 | **83.72 ± 0.18** |

some others in the former category for the ResNet-34 architecture and *CIFAR-10N* dataset. As can be seen in Table 11, outlier gradient analysis features in the top-2 performers compared to the other noisy learning baselines. We would like to emphasize that this is not an exhaustive list of baselines and noisy learning by adjusting the loss/model is not the primary focus of our work (but detecting detrimental samples is). Note that our algorithm could also be combined with approaches from this other category for additional gains.

### C.8. Comparison with GEX (Kim et al., 2024) and TRAK (Park et al., 2023)

We also compare the performance of our outlier gradient analysis methods with GEX (Kim et al., 2024) and TRAK (Park et al., 2023) , two new influence function methods. As mentioned in the main paper, we were able to obtain results for *CIFAR-10N* but obtained out-of-memory (OOM) errors for *CIFAR-100N*. This computational memory overhead highlights the shortcomings of these approaches. Furthermore, the results on *CIFAR-10N* for all three noise settings are shown in Table 12. As can be seen, our outlier gradient analysis approaches outperform these new influence function baselines for the detrimental data identification task.

### C.9. Experiments on Adapting Outlier Gradient Analysis to Validation/Test Set

We also conduct experiments for the distribution shift benchmark from the influence function work by (Chhabra et al., 2024). These experiments will showcase the applicability of outlier gradient analysis in adapting to a validation/test set distribution (instead of solely relying on the training set distribution). In (Chhabra et al., 2024), three distribution shift scenarios are considered on the *Folktables ACS-Income* (Ding et al., 2021) dataset: time-shifted, location-shifted, and time+location-shifted. Essentially, in each of these settings, either the train/test distribution are time-shifted (e.g. 2014/2018), location-shifted (e.g. CA/MI), or both (e.g. 2014 & CA / 2018 & MI). We undertake the same experiments but using the OneClassSVM semi-supervised outlier analysis approach (Li et al., 2003) instead of iForest, L1/L2 norm, and provide the test set as inliers to correct the distribution of the training set influence estimation. Then, we utilize outlier gradient analysis for each setting, with results shown in Table 13. Our approach is highly adaptable to differing test/validation set distributions (concept drift) and can significantly outperform other baselines in this setting as well.

*Table 12.* Performance comparison of our outlier gradient analysis methods with GEX (Kim et al., 2024) and TRAK (Park et al., 2023) influence function baselines.

| Method | CIFAR-10N (Aggregate) | CIFAR-10N (Random) | CIFAR-10N (Worst) |
|---|---|---|---|
| GEX (Kim et al., 2024) | 90.67 | 89.13 | 80.30 |
| TRAK (Park et al., 2023) | 91.73 | 90.07 | 83.52 |
| **Outlier Gradient (L1)** | 91.86 | **90.66** | **84.20** |
| **Outlier Gradient (L2)** | **92.21** | 90.25 | 82.99 |
| Outlier Gradient (iForest) | 91.36 | 90.20 | 83.72 |

*Table 13.* Using OneClassSVM as the outlier analysis approach in the distribution shift experiments of (Chhabra et al., 2024) on the *Folktables ACS-Income* dataset.

| Method | Time | Loc | Time + Loc |
|---|---|---|---|
| Gradient Tracing | 0.7523 | 0.7628 | 0.7483 |
| DataInf | 0.7390 | 0.7830 | 0.7547 |
| LiSSA | 0.7490 | 0.7657 | 0.7498 |
| Self-DataInf | 0.7783 | 0.7797 | 0.7812 |
| Self-LiSSA | 0.7782 | 0.7798 | 0.7782 |
| Outlier Gradient Analysis (L1) | 0.7683 | 0.7797 | 0.7742 |
| Outlier Gradient Analysis (L2) | 0.7687 | 0.7760 | 0.7690 |
| Outlier Gradient Analysis (iForest) | 0.7708 | 0.7892 | 0.7750 |
| **Outlier Gradient Analysis (OneClassSVM)** | **0.7765** | **0.8063** | **0.7840** |

# D. Code and Reproducibility

We provide our code, instructions, and implementation in an open-source repository: `https://github.com/anshuman23/outlier-gradient-analysis`. The experiments were conducted on two separate Linux (Ubuntu 20.04.6 LTS) servers– the experiments of Sections 6 and 7 were conducted on NVIDIA GeForce RTX A6000 GPUs with 50GB VRAM running CUDA version 12.0 and all other experiments were conducted on an NVIDIA Tesla V100 with 32GB VRAM and CUDA version 11.4.

