# OpenReview forum: "Outlier Gradient Analysis: Efficiently Identifying Detrimental Training Samples for Deep Learning Models"
_ICML.cc/2025/Conference — ICML 2025 oral_

### Official Review · Reviewer_MZJ6 · 2025-03-14

**Overall Recommendation:** 4

**Summary:**

This paper introduces a novel approach called "Outlier Gradient Analysis" for identifying detrimental training samples in deep learning models. The authors establish a conceptual bridge between influence functions (a traditional method for assessing training data impact) and outlier detection in the gradient space. The key innovation is transforming the computationally expensive task of calculating influence functions, which requires inverting the Hessian matrix, into a more efficient outlier detection problem in the gradient space.

**Claims And Evidence:**

The paper makes several key claims, which are generally well-supported by evidence:

1. **Detrimental samples can be identified as outliers in the gradient space**: The authors provide strong theoretical justification through Observation 3.1 and Hypothesis 3.2, establishing that detrimental samples are typically a minority in the training set and can be detected as outliers in the gradient space.

2. **Outlier Gradient Analysis is more computationally efficient than traditional influence functions**: The authors provide empirical evidence through running time comparisons (referenced in Section 8 and detailed in Appendix C.3), showing that their method is significantly faster than influence function approaches that require Hessian computation or approximation.

3. **Outlier Gradient Analysis performs competitively or better than existing methods**: This claim is well-supported through extensive experiments across multiple domains:
   - On synthetic datasets, the method achieves 96-98% accuracy in identifying mislabeled samples, outperforming all baselines (Table 1).
   - On CIFAR-10N and CIFAR-100N, the method consistently ranks among the top performers across different noise settings (Table 2).
   - On NLP fine-tuning tasks, the method outperforms all baselines on 3 out of 4 GLUE datasets and matches the best baseline on the fourth (Figure 3).
   - On LLM influential data identification, the method achieves perfect scores for both AUC and Recall metrics (Table 3).

**Essential References Not Discussed:**

N/A

**Experimental Designs Or Analyses:**

The experimental designs are thorough and well-executed. The authors evaluate their approach across a diverse range of applications, model architectures, and datasets, demonstrating its versatility and effectiveness.

**Methods And Evaluation Criteria:**

The methodology is sound and well-justified. The authors clearly explain the theoretical foundation of their approach, establishing the connection between influence functions and outlier detection in the gradient space. The transformation is elegantly formulated and the implementation details are thoroughly described.

The experimental setup is comprehensive, covering a diverse range of applications and model architectures. The authors also conduct ablation studies on key hyperparameters and provide running time analyses to demonstrate computational efficiency.

**Other Comments Or Suggestions:**

N/A

**Other Strengths And Weaknesses:**

1. Although this work conducted experiments on LLMs, I find it strange that they only used LLMs for classification tasks. It would make more sense to experiment with LLMs' generation tasks. This is my biggest concern, and if the authors can address this concern, I would be happy to raise the score.


2. Sensitivity to outlier detection algorithm. The performance of the method depends on the choice of outlier detection algorithm and its hyperparameters. While the paper explores different options (iForest, L1/L2-norm), a more systematic analysis of this dependency would be valuable.

**Questions For Authors:**

N/a

**Relation To Broader Scientific Literature:**

N/A

**Theoretical Claims:**

The paper makes several theoretical claims, primarily in Section 3, which establishes the connection between influence functions and outlier detection in the gradient space.

---

> ### Author Rebuttal · Authors · 2025-03-31
>
> Dear Reviewer MZJ6,
>
> Thank you for your efforts in reviewing our work, we are grateful for your insights. We provide answers to the questions raised, below:
>
> - **Although this work conducted experiments on LLMs, I find it strange that they only used LLMs for classification tasks. It would make more sense to experiment with LLMs' generation tasks. This is my biggest concern, and if the authors can address this concern, I would be happy to raise the score.**
>
>     Thank you. We would like to point out that the current LLM benchmarks are not classification tasks, but generation tasks. Each of the 3 influential data identification tasks (Math With Reasoning, Math Without Reasoning, and Sentence Transformations) we considered from past work (Kwon et al, 2024) have 10 classes/categories of subtasks (e.g. Sentence Transformation can have 10 different types of natural language transformations and Math problems can have 10 different categories of word problems) but are still generation tasks. More details regarding these benchmarks are provided in Appendix B.1.4 and B.1.5. In general, we agree with the reviewer that one of the challenges associated with influential data identification in LLMs is the lack of more complex and varied benchmarks. This is currently an evolving field, and designing benchmarks is challenging because ground-truth influence labels need to accurately reflect the model's inductive bias for a test sample (i.e. the model should find the training samples most influential for a particular test example). Past work does this by making the train-test problem sub-categories very similar. We are now working on designing better LLM influence identification benchmarks for future work, and hope to test our outlier gradient analysis methods on these benchmarks as well.
>
> ___
>
> - **Sensitivity to outlier detection algorithm. The performance of the method depends on the choice of outlier detection algorithm and its hyperparameters. While the paper explores different options (iForest, L1/L2-norm), a more systematic analysis of this dependency would be valuable.**
>
>      Thank you for the great question.
>
>      Regarding outlier analysis algorithms, in the paper, we have utilized 4 outlier analysis algorithms: iForest (main paper), L1 norm (main paper), L2 norm (main paper), and OneClassSVM (Appendix C.9) for outlier gradient analysis. For all 4 of these algorithms, the general trend of outlier gradient analysis improving model performance against competitive baselines in the noisy learning regime can be observed. Our primary aim in choosing these algorithms was their high computational efficiency and minimal number of hyperparameters.
>
>      Regarding hyperparameter sensitivity, we had provided details and additional experiments on hyperparameters in **Appendix C**, which we also discuss below. More specifically, our outlier analysis algorithm has two hyperparameters: (1) the trimming budget $k$ which is the number of samples to remove, and (2) the hyperparameters of the outlier detection algorithm being used (e.g. iForest).
>
>     1. For the first hyperparameter of the trimming budget $k$, we conduct additional experiments while varying the value of $k$ from 2.5% to 12.5% for all 4 noise settings of CIFAR-10N. These results are provided in **Table 5** of **Appendix C.2**.  As can be seen, the highest values across each noise regime are obtained by outlier gradient analysis (L2 norm thresholding at 12.5% for Aggregate and Random; and L2 norm thresholding at 2.5% for Worst), indicating its broad suitability. These results also show that the budget of 5% is a good choice for the trimming budget, leading to desirable performance in most cases.
>
>     2. The second hyperparameter choice is only a facet of the iForest outlier analysis algorithm and constitutes the number of trees being used in the algorithm. Note that this is because for the L1 and L2 norm thresholding approaches, we do not have a norm threshold that needs to be chosen manually, since setting the budget automatically decides the threshold. For the iForest algorithm, we have provided results for varying the number of tree estimators on CIFAR-10N in **Table 8** of **Appendix C.4**. As can be observed, the performance of outlier gradient analysis remains stable across the board when the number of trees/estimators are varied, indicating low sensitivity of this hyperparameter to final results obtained.
>
> ___
>
> Thank you once again for helping improve our paper, we appreciate it.

---

> > ### Comment · Reviewer_MZJ6 · 2025-04-07
> >
> > Thanks to the authors for their responses during the rebuttal period. I now have a deeper understanding of the details in the paper, and I will accordingly raise my score.

---

> > > ### Author Response · Authors · 2025-04-07
> > >
> > > Dear Reviewer MZJ6,
> > >
> > > Thank you for engaging with us and for all your efforts spent in reviewing our work, they are greatly appreciated.
> > >
> > > Regards,
> > >
> > > Authors.

---

### Official Review · Reviewer_r7gY · 2025-03-19

**Overall Recommendation:** 4

**Summary:**

This paper proposed Outlier Gradient Analysis, establishing a theoretical bridge between influence functions (a common tool for this task) and outlier detection in the gradient space. The key insight is that detrimental samples can be effectively identified as outliers in the gradient space without computing the Hessian matrix—a major computational bottleneck in traditional influence function approaches. The method employs Isolation Forest, L1-norm, and L2-norm thresholding for outlier detection and demonstrates strong performance across CV and NLP.

# **update after rebuttal**
I would like to thank the authors for their sincere efforts to address my concerns. I have increase my score.

**Claims And Evidence:**

The paper's claims are generally well-supported by empirical evidence.

1. Gradient outliers correlate strongly with detrimental samples, validated through synthetic datasets where ground truth is known (showing 96-98% detection accuracy).

2. The approach outperforms baseline methods on vision tasks.

3. Computational efficiency claims are substantiated with runtime measurements showing orders of magnitude speedup over traditional influence methods.

However, evidence could be strengthened:
- LLM influence task shows perfect scores (1.0 AUC/Recall), which may read about task difficulty or potential ceiling effects.

**Essential References Not Discussed:**

There is relevant research on sample importance/informative evaluation and data valuation that deserves discussion in relation to the authors' proposed method (influence functions). These fall into two main categories:

1. Data Valuation Methods:
These approaches provide alternative frameworks for quantifying sample importance:

[1] LAVA: Data Valuation without Pre-Specified Learning Algorithms, arXiv 2023.

[2] Data-OOB: Out-of-bag Estimate as a Simple and Efficient Data Value, ICML 2023.

[3] Data Banzhaf: A Robust Data Valuation Framework for Machine Learning, AISTAT 2023.

[4] Training data influence analysis and estimation: A survey, Machine Learning, 2024.

2. Data Pruning Methods:
These methods are relevant, while some of them directly address problems similar to identifying detrimental samples:

[5] Robust and Fully-Dynamic Coreset for Continuous-and-Bounded Learning (With Outliers) Problems, NeurIPS 2021.

[6] Robust Data Pruning under Label Noise via Maximizing Re-labeling Accuracy, NeurIPS 2023.

[7] Feature Distribution Matching by Optimal Transport for Effective and Robust Coreset Selection, AAAI, 2024.

[8] Instance-dependent Early Stopping, ICLR 2025.

**Experimental Designs Or Analyses:**

1. The synthetic experiments in Section 4 provide clear validation of the core hypothesis with controlled conditions.

2. The CIFAR-10N/100N experiments appropriately assess performance across varying noise regimes.

3. The ablation studies on trimming budget k and iForest parameters are valuable.

4. The computational complexity analysis is good enough.

That is enough, other minor issues are not concern me.

**Methods And Evaluation Criteria:**

1. The use of varied datasets spanning different domains (vision, NLP, LLMs) demonstrates generalizability.

2. In CV tasks, the evaluation protocol of detecting and removing detrimental samples followed by retraining is a sensible approximation of real-world application scenarios.

3. As a concern, I suggest that the authors expand their comparison beyond just Influence function baselines. In the area of learning with noisy labels, there are many mature sample selection methods with objectives similar to the proposed approach. For example, using the small-loss criterion from early-stopped models [1] to select clean samples, or leveraging whether models can consistently learn a sample as a criterion to identify mislabeled examples [2]. I recommend incorporating these methods into your comparison to evaluate both selection accuracy and computational efficiency.

[1] Co-teaching: Robust Training of Deep Neural Networks with Extremely Noisy Labels, NeurIPS 2018.

[2] Late Stopping: Avoiding Confidently Learning from Mislabeled Examples, ICCV 2023.

**Other Comments Or Suggestions:**

1. Moving table of runtime evaluation to the main text, which is a key contribution of the work.
2. Minor typos and formatting issues:
   - Line 116: "as a the discrete version" → "as the discrete version"

**Other Strengths And Weaknesses:**

**Strengths:**
1. The paper is a good read, and located everything on their way.
2. Tthe authors smartly narrowed down the problem. Instead of trying to calculate exactly how much each sample influences the model (like traditional methods do), they simply focus on identifying which samples are harmful. This simplification turns a complex calculation into a straightforward binary problem.
3. The proposed methd tested on CV/NLP/LLM, and it works.

**Weaknesses:**

Influence functions provide a richer quantification (in float) of each sample's importance, potentially enabling a wider range of applications beyond sample selection. In this sense, it might seem reasonable that these methods aren't directly comparable to simpler sample selection techniques.

However, While the authors cleverly narrowed down the problem to avoid the computational complexity of influence functions, this simplification create the need for broader comparisons.

The paper would be in stronger position if it included comparisons with: sample selection methods for learning with noisy labels and data pruning for training efficiency. These methods may have similar objectives (identifying samples to remove for different reason) but use different approaches.

**Questions For Authors:**

1. For the LLM influential data identification task, can iForest estimators scale to different tasks with many classes?

2. Have you explored using other outlier detection algorithms beyond iForest and L1/L2 norms?

3. The approach currently requires computing gradients for all training samples. Why not considered approximation or sampling technique?

**Relation To Broader Scientific Literature:**

This work relates to several research directions in the machine learning literature:

1. It extends influence function research by providing a more computationally efficient method for the specific task of detrimental sample identification.

2. It connects to the noisy label learning literature by offering an effective approach for identifying mislabeled samples.

3. It contributes to data-centric AI  by focusing on improving model performance through data quality rather than model architecture.

4. It relates to outlier detection literature (which I am not familiar).

The authors appropriately position their work within these research areas and have a decent realted work.

**Theoretical Claims:**

The paper primarily presents an intuitive conceptual transformation rather than formal theoretical proofs. The core theoretical claim is Hypothesis 3.2, which establishes a connection between detrimental sample identification and outlier detection in gradient space. This hypothesis is supported by empirical evidence rather than formal proof, which is reasonable given the nature of the problem.

The justification in Section 3.2 about why the gradient term should be decisive in determining whether a sample is detrimental is logical. The validity of Observation 3.1 (that detrimental samples are a minority in converged models) is crucial to the approach and appears empirically sound, though not rigorously proven.

---

> ### Author Rebuttal · Authors · 2025-03-31
>
> Dear Reviewer r7gY,
>
> Thank you for your thoughtful review and feedback, we appreciate it. We have answered questions raised, below:
>
> - **The paper would be in stronger position if it included comparisons with: sample selection methods for learning with noisy labels and data pruning for training efficiency**
>
>     Thank you for the great suggestion. Along with comparisons with other influence function baselines and simpler noisy label correction methods, we had compared with a number of sample selection methods designed for noisy learning in **Appendix C.7 (Table 11)**. Approaches for noisy learning can be categorized into (1) methods that either change the loss function or model architecture or (2) those that identify noisy samples and remove/relabel them for improving model performance (Algan & Ulusoy, 2021). While the main paper has results for the latter category, we compare with the former category on CIFAR-10N (all 3 noise settings) in Table 11 of Appendix C.7. As can be observed, outlier gradient analysis is the top performer across these methods as well. While we had thought of also comparing with training data efficiency methods, we could not undertake a fair comparison as several methods opt for reducing the size of the training set as much as possible while ensuring that performance on the reduced set remains as close to the original set. However, the goal in our work is to specify a small trimming budget and increase performance as much as possible, leading to the research questions between these approaches being fundamentally different.
>
> ___
>
> - **For the LLM influential data identification task, can iForest estimators scale to different tasks with many classes?**
>
>     Currently, each of the 3 influential data identification tasks (Math With Reasoning, Math Without Reasoning, and Sentence Transformations) we considered from past work (Kwon et al, 2024) each have 10 classes/categories of subtasks (e.g. Sentence Transformation can have 10 different types of natural language transformations and Math problems can have 10 different categories of word problems). More details regarding these benchmarks are provided in Appendix B.1.4 and B.1.5. While the reviewer makes a great suggestion of scaling to even more classes, one of the challenges associated with influential data identification in LLMs is the lack of more complex and varied benchmarks. This is currently an evolving field, and designing benchmarks is challenging because ground-truth influence labels need to accurately reflect the model's inductive bias for a test sample (i.e. the model should find the training samples most influential for a particular test example). Past work does this by making the train-test problem categories very similar. We are now working on designing better LLM influence identification benchmarks for future work, and hope to test our outlier gradient analysis methods on these benchmarks as well.
>
> ___
>
> - **Have you explored using other outlier detection algorithms beyond iForest and L1/L2 norms?**
>
>     In the paper, we have utilized 4 outlier analysis algorithms: iForest (main paper), L1 norm (main paper), L2 norm (main paper), and OneClassSVM (Appendix C.9) for outlier gradient analysis. For all 4 of these algorithms, the general trend of outlier gradient analysis improving model performance against competitive baselines in the noisy learning regime can be observed. Our primary aim in choosing these algorithms was their high computational efficiency and minimal number of hyperparameters.
>
> ___
>
> - **The approach currently requires computing gradients for all training samples. Why not consider approximation or sampling technique?**
>
>     The reason we did not opt for approximating or sampling gradients is the ease with which they are available for model deep learning models trained via backpropagation. Basically, we can obtain gradients in one-pass as the model is training post each backpropagation step. Owing to the ease of gradient access, we did not optimize this further via approximation. Moreover, approximation/sampling would lead to some reduction in performance as opposed to using the original first-order gradients. As an aside, the Hessian is not available during training since it contains second order information.
>
> ___
>
> - **Moving table of runtime evaluation to the main text and minor typos**:
>
>     Thank you for pointing these out. We will incorporate these suggestions into the revision as requested.
>
> ___
>
> Thank you once again for all your time and effort in reviewing our work, and helping strengthen our contributions.

---

> > ### Comment · Reviewer_r7gY · 2025-04-02
> >
> > I would like to thank the authors for their sincere efforts to address my concerns. I am inclined to increase my score by +1 (as a result, 4). I would like to seeing all the revisions regarding my comment in the updated version.

---

> > > ### Author Response · Authors · 2025-04-03
> > >
> > > Dear Reviewer r7gY,
> > >
> > > We are grateful for your engagement and are happy to hear that your concerns were addressed. We will definitely incorporate the suggested revisions in our paper as promised.
> > >
> > > Thank you once again for all your efforts, we appreciate it.
> > >
> > > Regards,
> > >
> > > Authors.

---

### Official Review · Reviewer_p9JH · 2025-03-21

**Overall Recommendation:** 4

**Summary:**

This paper addresses the challenge of identifying training samples that negatively impact deep learning model performance.  The authors draw a connection between identifying detrimental training samples using influence functions and outlier detection in the gradient space.  This connection leads to a Hessian-free formulation, reducing the computational cost associated with calculating the inverse of the Hessian matrix.
The authors propose an "outlier gradient analysis" approach.  They validate this approach on synthetic datasets and demonstrate its effectiveness in noisy label correction for vision models.  They also show its applicability to data selection for fine-tuning NLP models and influential data identification for Large Language Models (LLMs).

**Claims And Evidence:**

Claim 1: The paper builds a bridge between identifying detrimental training samples via influence functions and outlier detection on the gradient space of samples.


Evidence: The paper dedicates Section 3.2 to "Bridging Influence Estimation and Outlier Analysis," detailing the conceptual transformation from influence function-based detrimental sample identification to outlier detection in the gradient space.  Hypothesis 3.2 explicitly states the existence of outlier analysis algorithms for detecting detrimental samples in the gradient space.  The proposed outlier gradient analysis approach is then detailed in Section 3.3.


Assessment: The claim is well-supported. The paper provides a clear explanation and justification for this connection.

Claim 2: The "transformation features a straightforward and Hessian-free formulation, and reduces the computational cost associated with the Hessian matrix and its inverse."


Evidence: The paper emphasizes that outlier gradient analysis "not only features a straightforward and Hessian-free formulation"  but also "reduces the computational cost associated with the Hessian matrix and its inverse."  The method operates directly on the gradient space, avoiding the computation and inversion of the Hessian.  Algorithm 1 outlines the approach, which does not involve Hessian calculations.  Experiments in Section 8 discuss computational complexity and running time, showing that outlier gradient analysis is computationally efficient.  Table 7 provides a comparison of computational complexities, highlighting that outlier gradient analysis has a lower complexity than Hessian-based methods.


Assessment: The claim is convincingly supported by the presented methodology and experimental results.

In summary, the claims made in the submission are generally well-supported by the evidence provided.

**Essential References Not Discussed:**

Gradient-based anomaly detection: There is a body of work that directly uses gradients for anomaly detection, where anomalies are identified based on their gradient patterns. This line of work is closely related to the paper's idea of using gradients to identify outliers, and discussing it would provide a broader context.

References:

- Huang, Rui, Andrew Geng, and Yixuan Li. "On the importance of gradients for detecting distributional shifts in the wild." Advances in Neural Information Processing Systems 34 (2021): 677-689.
- Kwon, Gukyeong, et al. "Backpropagated gradient representations for anomaly detection." Computer Vision–ECCV 2020: 16th European Conference, Glasgow, UK, August 23–28, 2020, Proceedings, Part XXI 16. Springer International Publishing, 2020.
- Chen, Jinggang, et al. "Gaia: Delving into gradient-based attribution abnormality for out-of-distribution detection." Advances in Neural Information Processing Systems 36 (2023): 79946-79958.
- ElAraby, Mostafa, et al. "GROOD: GRadient-aware Out-Of-Distribution detection in interpolated manifolds." arXiv preprint arXiv:2312.14427 (2023).

**Experimental Designs Or Analyses:**

The experimental designs and analyses are sound and appropriate for evaluating the proposed method across different tasks and datasets.

**Methods And Evaluation Criteria:**

The authors propose "outlier gradient analysis," which connects the identification of detrimental samples using influence functions to outlier detection in the gradient space. This method is designed to address the computational challenges of traditional influence functions, which require Hessian matrix inversion. The choice of outlier detection algorithms like Isolation Forest is justified based on efficiency and effectiveness.

Evaluation Criteria: The paper uses a combination of synthetic datasets and real-world noisy label datasets (CIFAR-10N, CIFAR-100N). For the synthetic data, they measure ground-truth outlier predictive accuracy and performance gain. For real-world datasets, they evaluate the accuracy of noisy label correction. They also extend their evaluation to data selection for fine-tuning NLP models (GLUE datasets) and influential data identification for Large Language Models, using appropriate metrics (AUC and Recall).

Rationale: These evaluation choices are relevant because they cover a range of scenarios, from controlled synthetic environments to more complex real-world applications in computer vision and natural language processing. The use of noisy label datasets is particularly relevant to evaluating the method's ability to identify detrimental samples.

**Other Comments Or Suggestions:**

Further Analysis of Outlier Detection Algorithms: The paper justifies the use of Isolation Forest (iForest) but could benefit from a more detailed analysis and comparison of other outlier detection algorithms. This would provide a more comprehensive understanding of the impact of different outlier detection techniques on the proposed method.

**Other Strengths And Weaknesses:**

Strengths:

- Originality: The paper presents an original approach by connecting influence functions with outlier detection in the gradient space. This is a novel way to address the computational challenges of influence functions and offers a new perspective on identifying detrimental training samples. The idea of using outlier analysis for this purpose is creative and potentially impactful.

- Significance: The problem of identifying detrimental training samples is a significant challenge in data-centric learning. The proposed method has the potential to improve the efficiency and scalability of influence estimation, making it more applicable to large-scale deep learning models. This could have a substantial impact on various applications, including noisy label correction, data selection, and model interpretation.

- Clarity: The paper is generally well-written and the proposed method is explained clearly. The authors provide sufficient background information and motivation for their approach.

- Thorough Evaluation: The authors evaluate their method on synthetic datasets, noisy label correction for vision, data selection for NLP, and influential data identification for LLMs. This comprehensive evaluation demonstrates the broad applicability and effectiveness of the proposed approach.

Weaknesses:

- Limited Exploration of Outlier Detection Methods: While the paper justifies the use of Isolation Forest, it does not thoroughly explore or compare a wider range of outlier detection algorithms. There might be other more suitable algorithms that could further improve the performance or efficiency of the proposed method.

Lack of Discussion on Failure Cases: The paper could include a more detailed discussion of potential limitations and failure cases of the proposed method. Understanding when and why the method might not perform optimally is crucial for a comprehensive analysis.

**Questions For Authors:**

How sensitive is your method to the choice of hyperparameters, such as the number of trees in iForest or the threshold for L1/L2 norm methods? Please provide more guidance on how to choose appropriate hyperparameter values for different datasets and model architectures. A discussion on the robustness of the method to hyperparameter settings would be valuable.

**Relation To Broader Scientific Literature:**

The key contributions of this paper are related to the broader scientific literature in the following ways:

* **Influence Functions:** The paper builds upon the existing body of work on influence functions, a technique used for estimating the impact of training data on model predictions. It addresses the computational limitations of influence functions, specifically the high cost of inverting the Hessian matrix, which becomes a bottleneck for large-scale deep learning models.

* **Data-Centric Learning:** The research aligns with the growing field of data-centric learning, which focuses on improving model performance by manipulating the training data rather than the model architecture. The problem of identifying and removing detrimental samples is a core challenge in this area.

* **Outlier Detection:** The paper connects the problem of detrimental sample identification to the field of outlier detection. By framing detrimental samples as outliers in the gradient space, the authors leverage existing outlier detection algorithms to solve the problem of identifying harmful data points.

**Theoretical Claims:**

The primary theoretical claim revolves around "Hypothesis 3.2" and its connection to the proposed "outlier gradient analysis."

Hypothesis 3.2: "There exist outlier analysis algorithms capable of detecting detrimental samples in the gradient space."

The authors build a conceptual bridge between influence functions and outlier detection in gradient space.  They argue that detrimental samples, which negatively impact model utility, can be considered outliers in the gradient space.  This is supported by Observation 3.1, which states that in a converged model, most training samples contribute positively, while detrimental samples are a minority.

The paper doesn't provide formal proofs in the mathematical sense for Hypothesis 3.2. Instead, it offers a logical argument and empirical evidence to support it. The argument relies on the observation that detrimental samples are analogous to outliers and the justification that gradients play a decisive role in determining a sample's influence.

While the paper doesn't offer formal proofs, the logical reasoning and empirical results provide strong evidence for the validity of the central theoretical claim.

---

> ### Author Rebuttal · Authors · 2025-03-31
>
> Dear Reviewer p9JH,
>
> Thank you for your insightful review, comments, and suggestions. We answer the questions raised, below:
>
> - **Essential References Not Discussed**:
>
>      Thank you for providing these references on gradient-based outlier/anomaly detection, we appreciate it. As the reviewer correctly points out, these are closely related to our work in a technical sense (but explore different research questions), and we will be sure to include and discuss these in the revision.
>
> ___
>
> - **Lack of Discussion on Failure Cases**:
>
>     We can definitely aim to add more discussion on the limitations of our approach in the main paper. While outlier gradient analysis is useful in cases where training data can be noisy, it might not be as useful if the data is already very high quality and there are no outlying gradient samples. However, this might not be the case in the real-world unless some steps have already been taken to ensure high data quality. Furthermore, outlier analysis algorithms have a fundamental limitation of how to specify the budget for outlier detection, which is a non-trivial hyperparameter optimization problem. While this is a common problem with little consensus across the entire field of outlier analysis, our methods inherit this limitation as well (although our methods work well for different budget thresholds, as shown in additional experiments in the Appendix C.2).
>
> ___
>
> - **Further Analysis of Outlier Detection Algorithms**:
>
>     Thank you for this suggestion. In our paper, we have utilized 4 outlier analysis algorithms: iForest, L1 norm thresholding, L2 norm thresholding, and OneClassSVM (Appendix C.9) for outlier gradient analysis. For all 4 of these algorithms, the general trend of outlier gradient analysis improving model performance against competitive baselines in the noisy learning regime can be observed. Also, note that our primary aim in choosing these algorithms was their high computational efficiency and minimal number of hyperparameters. More complex approaches generally tend to be slower (e.g. those based on deep learning) and thus, wouldn't be as useful for outlier analysis of the gradient space. However, we are happy to include any other useful outlier detection methods that the reviewer would like us to.
> ___
>
> - **How sensitive is your method to the choice of hyperparameters, such as the number of trees in iForest or the threshold for L1/L2 norm methods? Please provide more guidance on how to choose appropriate hyperparameter values for different datasets and model architectures. A discussion on the robustness of the method to hyperparameter settings would be valuable.**
>
>     Thank you for the great question. We had provided details and additional experiments on hyperparameters in **Appendix C**, which we also discuss below. More specifically, our outlier analysis algorithm has two hyperparameters: (1) the trimming budget $k$ which is the number of samples to remove, and (2) the hyperparameters of the outlier detection algorithm being used (e.g. iForest).
>
>     1. For the first hyperparameter of the trimming budget $k$, we conduct additional experiments while varying the value of $k$ from 2.5% to 12.5% for all 4 noise settings of CIFAR-10N. These results are provided in **Table 5** of **Appendix C.2**.  As can be seen, the highest values across each noise regime are obtained by outlier gradient analysis (L2 norm thresholding at 12.5% for Aggregate and Random; and L2 norm thresholding at 2.5% for Worst), indicating its broad suitability. These results also show that the budget of 5% is a good choice for the trimming budget, leading to desirable performance in most cases.
>
>     2. The second hyperparameter choice is only a facet of the iForest outlier analysis algorithm and constitutes the number of trees (as the reviewer correctly pointed out). Note that this is because for the L1 and L2 norm thresholding approaches, we do not have a norm threshold that needs to be chosen manually, since setting the budget automatically decides the threshold. For the iForest algorithm, we have provided results for varying the number of tree estimators on CIFAR-10N in **Table 8** of **Appendix C.4**. As can be observed, the performance of outlier gradient analysis remains stable across the board when the number of trees/estimators are varied, indicating low sensitivity of this hyperparameter to final results obtained.
>
> ___
>
> We would like to thank the reviewer again for all the time and effort spent on the review, and helping improve our work.

---

### Official Review · Reviewer_91NL · 2025-03-21

**Overall Recommendation:** 3

**Summary:**

The paper introduces a simple yet powerful alternative to traditional influence functions by leveraging outlier detection in gradient space. This method—Outlier Gradient Analysis—provides a scalable, efficient, and accurate way to identify harmful training samples, with broad utility across diverse deep learning domains.  Extensive experiments demonstrate the method’s good performance in both accuracy and computational efficiency.

**Claims And Evidence:**

**Claim 1: The majority of training samples positively contribute to the model’s utility,and a much smaller subset than beneficial samples**
Although this conclusion is evident, it would be more rigorous if there are quantitative results to support this claim.
**Claim 2: Gradient-space outliers correspond to detrimental training samples.**
The authors validate this key hypothesis on both **synthetic datasets** (linear and non-linear) and show that detrimental samples are clearly separable in gradient space. Due to the absence of theoretical equivalence proof, this key hypothesis should be validated on **real datasets**.
**Claim 3: Outlier Gradient Analysis is a computationally efficient and outperforms other baselines.**
Experiments are provided across domains including CIFAR-10N/100N and LLM benchmarks.

**Essential References Not Discussed:**

None

**Experimental Designs Or Analyses:**

The experimental designs and analyses in the paper are generally sound and provide strong evidence for the effectiveness of the proposed Outlier Gradient Analysis method. The authors carefully design experiments across multiple domains (synthetic, vision, NLP, LLMs) and use appropriate evaluation metrics.

**Methods And Evaluation Criteria:**

The method addresses a significant gap in existing approaches. Influence functions, while effective, are computationally expensive due to the need for Hessian matrix inversion, especially in deep models. By shifting to gradient-space outlier detection, the authors propose an approach that avoids this bottleneck, making it both more scalable and efficient. This transformation from influence functions to outlier analysis is logically justified, and the use of simple and efficient outlier detection algorithms. However, this transformation requires more comprehensive validation on real datasets, or the authors should provide theoretical proof.

**Other Comments Or Suggestions:**

I have no other comments and suggestions.

**Other Strengths And Weaknesses:**

**Strengths**
The method is simple and efficient.
The idea of avoiding computing the Hessian matrix is innovative.
**Weaknesses**
The technical details have not been adequately elaborated.
The article lacks a discussion on the limitations of the methodology.

**Questions For Authors:**

**Q1: Could you elaborate on the computational specifics of Outlier Gradient (L1) and Outlier Gradient (L2)?**
**Q2: Why is there an additional summation symbol in Equation 1?**
**Q3: Avoiding the computation of the Hessian matrix can accelerate calculations, but it is strange that only calculating the third term of Equation 1 improves accuracy.**

**Relation To Broader Scientific Literature:**

None

**Theoretical Claims:**

This paper does not furnish the theoretical claims.

---

> ### Author Rebuttal · Authors · 2025-03-31
>
> Dear Reviewer 91NL,
>
> Thank you for your insightful review and feedback. We answer the questions raised, below:
>
> - **Q1: Could you elaborate on the computational specifics of Outlier Gradient (L1) and Outlier Gradient (L2)?**
>
>     Thank you for your question. Our outlier gradient approach (Algorithm 1) takes in as input a trimming budget $k$ and an outlier analysis algorithm $\mathcal{A}$. Here, $\mathcal{A}$ can be any outlier analysis algorithm, including L1 and L2 norm thresholding. For these two algorithms, we compute the gradient of the training samples, and then simply select the top-$k$% (i.e. based on budget) norm values as outliers. For L1 (or L2) norm thresholding, the L1 (or L2) norm values need to be computed, but this is a computationally efficient simple tensor operation.
>
> ___
>
> - **Q2: Why is there an additional summation symbol in Equation 1?**
>
>     The additional summation symbol is simply aggregating the loss for each of the validation/training set samples (i.e. samples from either $V$ or $T$) on which training sample $z_j$'s influence is being measured. As the loss is additive, the derivative of the loss is also additive, and hence, can be summed over to compute the overall loss contributions made by the individual training sample ($z_j$) we are computing influence for. This summation over the loss is used in past work on influence function for model performance measurement, such as [1,2], among others.
>
> ___
>
> - **Q3: Avoiding the computation of the Hessian matrix can accelerate calculations, but it is strange that only calculating the third term of Equation 1 improves accuracy**
>
>     Thank you for the great question. One intuitive reason for this (that also past work has found) is that the Hessian is not mandatory for influence analysis in all cases. While (Koh and Liang, 2017) pioneered influence functions based on the Hessian, other work has considered influence functions without relying on the Hessian such as TracIn [3], TracIn-Last [4], VAE-TracIn [5], BoostIn [6], Hydra [7], etc. Note that TracIn here is the Gradient Tracing baseline we compare with in the paper as well. TracIn and its variant approaches are simply calculating a vector inner product on the gradient space without computing the Hessian. Our outlier gradient analysis approach undertakes the Hessian-free in a novel manner by discovering detrimental training samples using the outlyingness of the gradient terms.
> ___
>
> - **The key hypothesis should be validated on real datasets**:
>
>     For real-world datasets (with a large number of classes) and large models, visualizing the gradient space (which will be very high-dimensional) requires undertaking aggressive approximations and is not possible in 2D/3D without losing useful information that describes outlyingness. Thus, we visualized the gradient space for simpler models and synthetic datasets as these are controllable and allow us to demonstrate our hypothesis, while real-world datasets could be affected by unknown factors. However, we would like to emphasize that the success of our methods on downstream real-world datasets (all the experiments in our paper) showcases the benefits and efficacy of outlier gradient analysis (thereby validating it) on real-world datasets and large models as well.
>
> ___
>
> - **The article lacks a discussion on the limitations of the methodology**:
>
>     Thank you for the suggestion. We can definitely aim to add more discussion on the limitations of our approach in the main paper. While outlier gradient analysis is useful in cases where training data can be noisy, it might not be as useful if the data is already very high quality and there are no outlying gradient samples. However, this might not be the case in the real-world unless some steps have already been taken to ensure high data quality. Furthermore, outlier analysis algorithms have a fundamental limitation of how to specify the budget for outlier detection, which is a non-trivial hyperparameter optimization problem. While this is a common problem with little consensus across the entire field of outlier analysis, our methods inherit this limitation as well (although our methods work well for different budget thresholds, as shown in additional experiments in Appendix C.2).
>
> ___
>
> Thank you once again for your help in strengthening our paper, we appreciate it.
> ___
>
> **References**:
> 1. Understanding black-box predictions via influence functions. ICML 2017.
> 2. DataInf: Efficiently Estimating Data Influence in LoRA-tuned LLMs and Diffusion Models. ICLR 2024.
> 3. Estimating training data influence by tracing gradient descent. NeurIPS 2020.
> 4. First is better than last for language data influence. NeurIPS 2022.
> 5. Understanding instance-based interpretability of variational auto-encoders. NeurIPS 2021.
> 6. Adapting and evaluating influence-estimation methods for gradient-boosted decision trees. JMLR 2023.
> 7. Hydra: Hypergradient data relevance analysis for interpreting deep neural networks. AAAI 2021.

---

> > ### Comment · Reviewer_91NL · 2025-04-02
> >
> > Thank you for your response. Most of my concerns have been addressed, except for the point that the key hypothesis should be validated on real datasets. Why can't high-dimensional data be projected into a low-dimensional space for visualization? Although the results may not be as perfect as those on synthetic data, it should still roughly validate the hypothesis of the paper. If it is not validated on real datasets, how can you prove that the results on downstream real data can be attributed to the hypothesis you proposed? After all, there is a significant gap between real datasets and synthetic data.

---

> > > ### Author Response · Authors · 2025-04-03
> > >
> > > Dear Reviewer 91NL,
> > >
> > > We would like to thank you for engaging with us and helping improve our contributions. As requested, we have undertaken additional experiments on two of our NLP datasets (SST2 and QNLI with RoBERTa as the model) to discuss the validity of our outlier gradient hypothesis on real datasets with detrimental/noisy samples:
> > >
> > > - For both the SST2 and QNLI datasets, we take the full gradient space (with 2048 dimensions) and apply iForest for outlier analysis of this gradient space. **We find that for (a) SST2, iForest detects noisy/detrimental training samples with an accuracy of _90.11%_ and for (b) QNLI, iForest detects noisy/detrimental training samples with an accuracy of _85.55%_. For both these real-world datasets, we can observe that our hypothesis is validated, as detected outliers correspond very highly to whether a training sample is noisy/detrimental or not.**
> > > ___
> > > - We had also stated in our rebuttal above that dimensionality reduction from the full gradient space to 2D or 3D might lose important outlyingness information. To validate this, we take the full 2048 dimensional gradient space for both datasets and reduce dimensionality using PCA. For ease of visualization, we randomly sample 100 samples and plot the top-2 PCA components for **SST2 (provided here: https://anonymous.4open.science/r/icml-rebuttal-2025/2d_grad_sst2.png)** and for **QNLI (provided here: https://anonymous.4open.science/r/icml-rebuttal-2025/2d_grad_qnli.png)**. The legend also shows whether a sample is noisy or not. **As can be observed from these figures, it is not possible to either algorithmically or manually detect outliers for such a low dimensional gradient space. Thus, the key takeaway here is that outlyingness might not be observed after aggressive dimensionality reduction, even with high outlier detection performance on the original high-dimensional gradient space.**
> > >
> > > ___
> > >
> > > We hope these additional experiments alleviate your concerns. Thank you once again for your efforts; we are grateful.
> > >
> > > Regards,
> > >
> > > Authors.

---

### Decision · Program_Chairs · 2025-05-01

**Decision:**

Accept (oral)

**Comment:**

This paper proposes an efficient method to identify detrimental training samples by detecting outliers in gradient space, thereby offering a Hessian-free alternative to traditional influence functions.
This submission received strong support from multiple reviewers:
- Reviewer p9JH praised the originality and computational efficiency of the proposed method.
- Reviewer r7gY highlighted the clarity of problem scoping and its broad applicability across CV, NLP, and LLM tasks.
-  Reviewer MZJ6 noted the elegance of the transformation and robustness of empirical results.
- Reviewer 91NL appreciated the simple yet powerful design and efficiency of the method.

Before the rebuttal, Reviewers raised concerns about the lack of validation of the key hypothesis on real datasets, limited comparison with methods from noisy label learning and data pruning, and unclear applicability to generation tasks in LLMs. These concerns were thoroughly addressed by the authors through additional experiments (e.g., iForest validation on SST2 and QNLI), clarifications on benchmarks used in LLMs, and further discussions in the rebuttal and appendix. Reviewers acknowledged and appreciated the authors’ responses.
Therefore AC recommends acceptance of the paper.